# Genome-wide association study of lung adenocarcinoma in East Asia and comparison with a European population

Lung adenocarcinoma is the most common type of lung cancer. Known risk variants explain only a small fraction of lung adenocarcinoma heritability. Here, we conducted a two-stage genome-wide association study of lung adenocarcinoma of East Asian ancestry (21,658 cases and 150,676 controls; 54.5% never-smokers) and identified 12 novel susceptibility variants, bringing the total number to 28 at 25 independent loci. Transcriptome-wide association analyses together with colocalization studies using a Taiwanese lung expression quantitative trait loci dataset ($n = 115$) identified novel candidate genes, including *FADS1* at 11q12 and *ELF5* at 11p13. In a multi-ancestry meta-analysis of East Asian and European studies, four loci were identified at 2p11, 4q32, 16q23, and 18q12. At the same time, most of our findings in East Asian populations showed no evidence of association in European populations. In our studies drawn from East Asian populations, a polygenic risk score based on the 25 loci had a stronger association in never-smokers vs. individuals with a history of smoking ($P_{interaction} = 0.0058$). These findings provide new insights into the etiology of lung adenocarcinoma in individuals from East Asian populations, which could be important in developing translational applications.

Lung adenocarcinoma (LUAD) is the most common histologic subtype of lung cancer and accounts for approximately 40% of lung cancer incidence worldwide[1–3]. In studies drawn from East Asian (EA) ancestry, LUAD has been the predominant histologic subtype among females[2] and has replaced squamous cell carcinoma as the most common subtype in males[4,5]. Well established risk factors, namely, tobacco smoking, certain environmental/occupational exposures and lifestyle factors, and family history, contribute to the risk of LUAD[6–8]. In addition, multiple genome-wide association studies (GWAS) have identified at least 24 susceptibility loci for LUAD that achieved genome-wide significance, many drawn from studies in EA[9–15] and European (EUR)[16–23] populations, as well as multi-ancestry meta-analyses[24,25]. Of these, 12 loci have been reported at genome-wide significance in GWAS of either never-smokers[9,11–13] or smokers and nonsmokers combined[10,14,15,24] in EA populations while another two loci were suggested in a multi-ancestry meta-analysis[24]. We estimated that the known susceptibility variants account for only 13% of the estimated familial risk in EA populations.

Accordingly, larger studies are needed to investigate the underlying architecture of susceptibility to LUAD in never-smokers and individuals with a history of smoking and in different ancestral populations. The importance of multi-ancestry analyses is further highlighted by reports of susceptibility loci showing association for LUAD in EA but not in EUR populations[13].

In the current study, we conducted a two-stage GWAS meta-analysis in EA populations using unpublished and previously published data from four studies: the Female Lung Cancer Consortium in Asia (FLCCA), Nanjing Lung Cancer Study (NJLCS)[10,24], National Cancer Center Research Institute (NCC) and Aichi Cancer Center (ACC), with 11,753 cases and 30,562 controls in the discovery set and 9905 cases and 120,114 controls in the replication set. A multi-ancestry meta-analysis of EA and EUR studies[16,22] (from the International Lung Cancer Consortium, ILCCO) was performed to identify variants shared by both populations. We also investigated the heterogeneity of effect sizes for susceptibility variants identified in EA and EUR populations[16,22] and

✉ e-mail: jianxin.shi@nih.gov; qingl@mail.nih.gov

obtained genome-wide estimates of effect-size correlation. Finally, we evaluated the genetic architecture[26] of LUAD, characterized by the number of susceptibility variants and their effect size distribution after normalizing allele frequencies, to investigate the accuracy of genetic risk prediction in the future GWAS in EA populations with increased sample sizes.

## Results

### Two-stage GWAS meta-analysis of LUAD in East Asian populations

For the discovery set, we performed a fixed-effect meta-analysis (11,753 cases and 30,562 controls) drawn from EA studies (Table 1, Supplementary Table 1). Details of quality control, imputation and post-imputation filtering are described in Methods. Variants with an imputation quality score ≥0.5 and minor allele frequency (MAF) ≥ 0.01 were included for meta-analysis. The estimated genetic correlation between LUAD in never-smokers and individuals with a history of smoking was rg = 0.81 (s.e. = 0.16) using linkage disequilibrium (LD) score regression (LDSC)[27], which enabled the primary meta-analysis to include the two groups. LDSC analysis suggested little evidence of residual population stratification (LDSC intercept = 1.03). We identified 14 loci achieving genome-wide significance $P < 5 \times 10^{-8}$ (Supplementary Table 2); two were novel at 2p23.3 (rs682888, OR = 0.89, $P = 4.94 \times 10^{-10}$) and at 7q31.33 (rs4268071, OR = 1.39, $P = 7.27 \times 10^{-10}$). In meta-analysis performed separately for males and females, and for never-smokers and individuals with a history of smoking, no further loci achieved genome-wide significance.

In the replication phase, we selected 38 lead variants with $P < 10^{-5}$ in the discovery data that were not previously reported as genome-wide significant in either EA or EUR populations and genotyped them in an independent data set of 9905 LUAD cases and 120,114 controls from a Japanese population (Table 1, Supplementary Table 1). After combining the discovery and the replication data, we identified a total of 10 novel loci achieving genome-wide significance and a novel variant on the locus at 15q21.2 that was previously reported in EUR populations[16] (Table 2, Manhattan plot in Fig. 1, and regional association plots in Supplementary Fig. 1).

Conditional analysis using GWAS summary statistics suggested two additional susceptibility variants rs13167280 (OR = 1.29, $P = 4.07 \times 10^{-13}$) and rs62332591 (OR = 0.87, $P = 3.21 \times 10^{-8}$) in the locus at 5p15.33 (Table 3, Supplementary Fig. 2); both are in modest LD with previously reported secondary variants in EA populations[28] ($R^2 = 0.27$ between rs13167280 and rs10054203;[28] $R^2 = 0.19$ between rs62332591 and rs10054203[28]). Variant rs12664490 (OR = 0.81, $P = 1.24 \times 10^{-10}$) was conditionally significant in a locus previously reported in EA at 6p21.1 (Table 3, Supplementary Fig. 3), adding another novel variant (12 novel variants in total).

A previous multi-ancestry meta-analysis conducted by Dai et al.[24] that included Chinese samples and EUR samples from the ILCCO study identified three SNPs for LUAD, one of which achieved genome-wide significance and the other two were suggestive in their analysis restricted to the Chinese subgroup[24] (see Supplementary Table 3). In the meta-analysis of the Chinese samples in Dai et al.[24]. with our independent EA samples, all three variants exceeded the threshold of genome-wide significance without issues of heterogeneity (Supplementary Table 3).

Overall, our study identified 12 novel susceptibility variants bringing the total to 28 genetic variants at 25 loci that have been identified to date in EA populations (Supplementary Table 4, Fig. 1). Assuming a familial risk estimate of 1.84 for first-degree relatives[29], the 25 independent susceptibility variants for LUAD (Supplementary Table 4) captured 16.2% of the familial relative risk in EA populations. Moreover, we found no evidence that the SNP associations differed between the samples from the Mainland of China and those from outside of the Mainland of China, or between Han Chinese and Japanese, the two largest ancestry populations in our study (Supplementary Table 5).

We further examined whether the novel variants identified in this study were associated with smoking behaviors (i.e., smoking status, cigarettes per day, initiation age and cessation) or chronic obstructive pulmonary disease in the Biobank Japan Project[30] (BBJ). We found no evidence that these variants were implicated in these traits in this cohort (Supplementary Table 6). A previous GWAS in EUR populations found variants (e.g., rs55781567) at the 15q25.1 CHRNA5 locus associated with tobacco smoking and lung cancer risk only in individuals with a history of smoking (OR = 1.33, $P = 1.83 \times 10^{-78}$, MAF = 0.39)[16,18,19,31]. However, this variant did not achieve genome-wide significance in our EA data (OR = 1.37, $P = 0.001$ for individuals with a history of smoking; OR = 1.05, $P = 0.44$ for never-smokers), likely because of a low MAF = 0.03, and no other variant in LD with this SNP showed a substantial association.

### Fine mapping and functional analyses of GWAS loci

To prioritize candidate variants for functional follow-up from each of the LUAD GWAS loci, we performed Bayesian fine mapping using FINEMAP[32] (Methods). Fine mapping of the genome-wide significant loci from the discovery set nominated 95% credible set variants for 9 loci with a median of 63 variants per locus (Supplementary Data 1). For the 12 novel variants identified from the combined discovery and replication datasets as well as conditional analysis, we then performed variant annotation analysis. High-LD variants for these signals ($R^2 \geq 0.8$ with the lead SNP in the 1000 Genomes, phase 3, EA) included those located in predicted promoters or enhancers in lung tissues/cells (RegulomeDB[33], Haploreg[34] v4.1, and FORGE2;[35] Supplementary Data 2), which can be tested in future experimental studies.

To further characterize the functionality of the prioritized susceptibility genes that could explain the new GWAS loci, eQTL colocalization and transcriptome-wide association study (TWAS) analyses

**Table 1 | Demographic characteristics of the subjects in the discovery and the replication datasets for a GWAS of lung adenocarcinoma in East Asians**

| | Discovery[a] | | Replication[b] | | Combined | |
|---|---|---|---|---|---|---|
| | **Cases** | **Controls** | **Cases** | **Controls** | **Cases** | **Controls** |
| Male | 4021 (34%) | 11,609 (38%) | 5650 (57%) | 62,596 (52%) | 9671 (45%) | 74,205 (49%) |
| Female | 7732 (66%) | 18,953 (62%) | 4255 (43%) | 57,518 (48%) | 11,987 (55%) | 76,471 (51%) |
| Individuals with smoking history | 3751 (32%) | 9780 (32%) | 6108 (62%) | 58,430 (49%) | 9859 (46%) | 68,210 (45%) |
| Never-smokers | 8002 (68%) | 20,782 (68%) | 3797 (38%) | 61,684 (51%) | 11,799 (54%) | 82,466 (55%) |
| Total | 11,753 | 30,562 | 9905 | 120,114 | 21,658 | 150,676 |

[a]The discovery dataset includes 4438 cases and 4544 controls from the FLCCA study, 1923 cases and 3544 controls from the NJLCS study, 3921 cases and 19,910 controls from the NCC study and 1471 cases and 2564 controls from the ACC study.

[b]The replication dataset consists of new candidate variant genotyping conducted in Japanese study LUAD subjects by the NCC study center and controls from the BioBank Japan. More details can be found in Supplementary Table 1 and Methods.

**Table 2 | Novel genetic variants associated with lung adenocarcinoma in East Asians**

| Chr | BP | SNP | Genes | Eff/Ref | EAF | Discovery | | Replication | | Combined | |
|---|---|---|---|---|---|---|---|---|---|---|---|
| | | | | | | OR (95% CI) | P | OR (95% CI) | P | OR (95% CI) | P |
| 3 | 138570011 | rs137884934 | *PIK3CB* | T/C | 0.09 | 0.81(0.74,0.89) | $6.33 \times 10^{-6}$ | 0.80(0.76,0.85) | $1.88 \times 10^{-15}$ | 0.80(0.77,0.84) | $6.21 \times 10^{-20}$ |
| 2 | 25757709 | rs682888 | *DTNB* | C/T | 0.47 | 0.89(0.86,0.93) | $4.94 \times 10^{-10}$ | 0.91(0.88,0.94) | $1.57 \times 10^{-10}$ | 0.90(0.88,0.92) | $5.96 \times 10^{-19}$ |
| 11 | 61581656 | rs174559 | *FADS1* | A/G | 0.39 | 0.91(0.88,0.94) | $6.10 \times 10^{-7}$ | 0.91(0.89,0.94) | $6.22 \times 10^{-9}$ | 0.91(0.89,0.93) | $1.93 \times 10^{-14}$ |
| 15 | 49757466 | rs71467682[a] | *FGF7, SECISBP2L* | G/A | 0.31 | 0.91(0.87,0.95) | $2.46 \times 10^{-6}$ | 0.90(0.88,0.93) | $2.30 \times 10^{-9}$ | 0.91(0.88,0.93) | $2.81 \times 10^{-14}$ |
| 10 | 126324209 | rs10901793 | *FAM53B, METTL10* | A/G | 0.30 | 1.10(1.06,1.14) | $3.14 \times 10^{-7}$ | 1.07(1.04,1.10) | $1.03 \times 10^{-5}$ | 1.08(1.06,1.11) | $3.04 \times 10^{-11}$ |
| 7 | 124373384 | rs4268071[b] | *GPR37* | T/G | 0.04 | 1.39(1.25,1.54) | $7.27 \times 10^{-10}$ | NA | NA | 1.39(1.25,1.54) | $7.27 \times 10^{-10}$ |
| 6 | 53389995 | rs531557 | *GCLC* | T/A | 0.60 | 0.90(0.87,0.94) | $7.73 \times 10^{-7}$ | 0.94(0.91,0.97) | $8.49 \times 10^{-5}$ | 0.93(0.90,0.95) | $9.25 \times 10^{-10}$ |
| 19 | 725066 | rs116863980 | *PALM* | A/G | 0.06 | 1.31(1.16,1.47) | $7.94 \times 10^{-6}$ | 1.17(1.09,1.26) | $2.50 \times 10^{-5}$ | 1.21(1.14,1.29) | $2.63 \times 10^{-9}$ |
| 15 | 56454223 | rs764014 | *RFX7* | G/A | 0.47 | 0.91(0.88,0.95) | $5.75 \times 10^{-7}$ | 0.95(0.92,0.98) | $7.36 \times 10^{-4}$ | 0.94(0.91,0.96) | $7.73 \times 10^{-9}$ |
| 4 | 44174404 | rs117715768 | *KCTD8* | T/C | 0.06 | 1.24(1.14,1.34) | $4.48 \times 10^{-7}$ | 1.10(1.04,1.17) | $1.28 \times 10^{-3}$ | 1.15(1.09,1.21) | $2.45 \times 10^{-8}$ |
| 4 | 157894892 | rs1373058 | *PDGFC* | A/T | 0.57 | 1.10(1.05,1.15) | $8.55 \times 10^{-6}$ | 1.06(1.03,1.09) | $3.60 \times 10^{-4}$ | 1.07(1.05,1.10) | $3.86 \times 10^{-8}$ |

All p values are nominal and two-sided.
[a]rs71467682 is in weak LD with rs77468143 ($R^2 = 0.27$ in EA) that was previously reported to be associated with LUAD in EUR populations[16].
[b]Replication data not available.

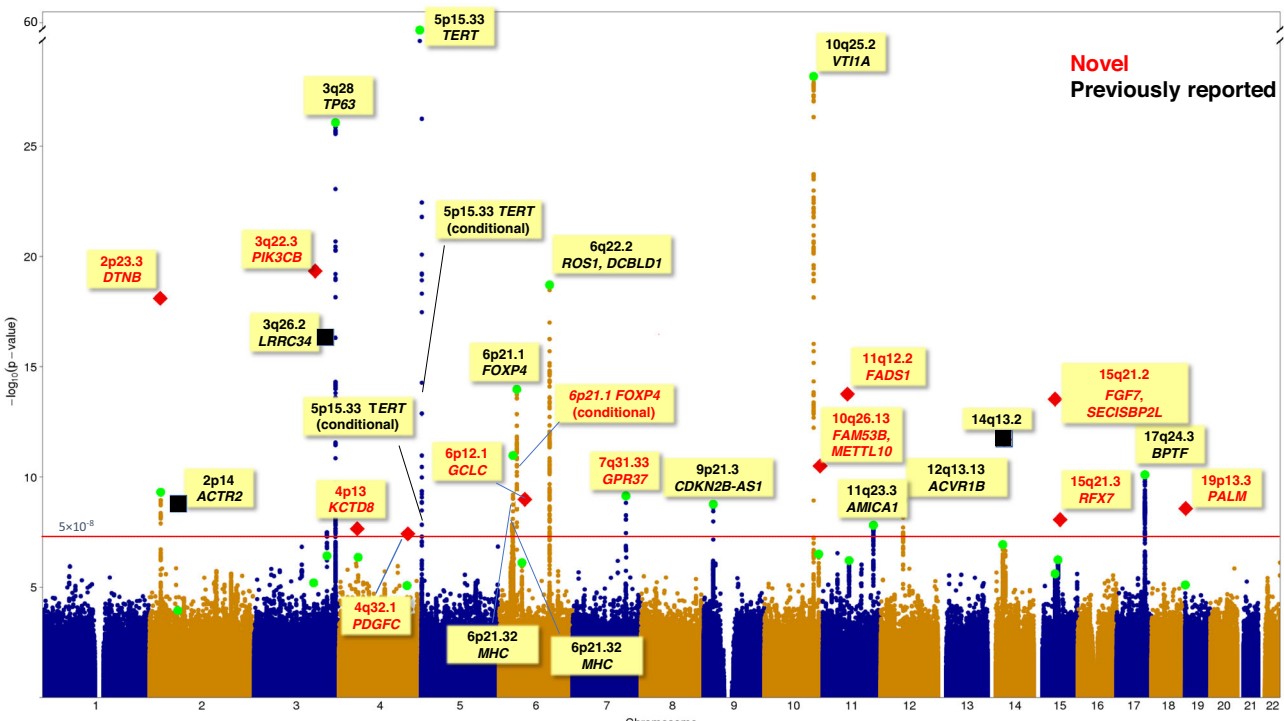

**Fig. 1 | Manhattan plot for GWAS meta-analysis of lung adenocarcinoma in East Asians.** The *x*-axis represents chromosomal location, and the *y*-axis represents -log₁₀(p-value). All *p* values were two-sided and not adjusted for multiple testing. The red horizontal line denotes the *p* value threshold for declaring genome-wide significance at $5 \times 10^{-8}$. For each box, red text represents a novel variant (12 novel variants, including the lead variants from 10 novel loci, rs12664490 by conditional analysis at 6p21.1, a locus previously reported in East Asians, and rs71467682 at 15q21.2, a locus preciously reported in Europeans); black text represents a previously reported association (16 variants in total, including three independently associated variants in 5p15.33 locus). For each locus, a green circle represents the top *p* value from the discovery samples, a red diamond represents the *p* value combining the discovery and the replication data, a black square represents the *p* value combining our discovery data and Chinese samples in Dai et al.[24] (for three variants identified in a cross-ancestry analysis of East Asians and Europeans in Dai et al.[24], see Supplementary Table 3). In summary, 28 variants at 25 loci achieved genome-wide significance, including 16 previously reported variants and 12 novel variants.

were conducted. Initial stratified LD score regression[36] using GTEx data (Supplementary Fig. 4; Supplementary Data 3) indicated that LUAD heritability drawn from EA populations are enriched in lung tissue-specific genes and chromatin features compared to other tissues (aggregated rank test $P = 1.36 \times 10^{-2}$ and $7.7 \times 10^{-3}$, respectively; Supplementary Data 3). Accordingly, we performed eQTL analyses using the Taiwanese dataset of adjacent normal lung tissues from 115 never-

smoking lung cancer patients (LCTCNS) (Methods; Supplementary data 4). We performed colocalization analyses of eQTL genes using eCAVIAR[37] and HyPrColoc[38]. A notable finding was the colocalization of *FADS1* at 11q12.2 (rs174559, posterior probability = 0.91) (Fig. 2; Supplementary Data 5), particularly since rs174559 was in LD with a recently identified functional variant (rs174557) regulating allelic *FADS1* expression in liver cells[39]. *FADS1* encodes fatty acid desaturase 1,

**Table 3 | Conditional and joint analyses identified independently associated risk SNPs for lung adenocarcinoma at two existing loci in East Asians**

| Chr | BP | SNP | Gene | GWAS analysis[a] | | | | Conditional analysis[b] | | Joint analysis[c] | |
|---|---|---|---|---|---|---|---|---|---|---|---|
| | | | | Eff/Ref | EAF | OR (95% CI) | P | OR (95% CI) | P | OR (95% CI) | P |
| 5 | 1280477 | rs13167280 | TERT | A/G | 0.22 | 1.47(1.37,1.57) | $6.99 \times 10^{-30}$ | 1.33(1.24,1.42) | $8.36 \times 10^{-17}$ | 1.29(1.20,1.38) | $4.07 \times 10^{-13}$ |
| 5 | 1286516 | rs2736100 | | A/G | 0.56 | 0.75(0.72,0.77) | $7.92 \times 10^{-58}$ | | | 0.80(0.77,0.83) | $9.83 \times 10^{-32}$ |
| 5 | 1290319 | rs62332591 | | G/T | 0.52 | 0.79(0.75,0.83) | $3.53 \times 10^{-23}$ | 0.87(0.83,0.91) | $2.95 \times 10^{-9}$ | 0.87(0.83,0.92) | $3.21 \times 10^{-8}$ |
| 6 | 41483390 | rs9367106 | FOXP4 | C/G | 0.32 | 1.20(1.15,1.26) | $1.06 \times 10^{-14}$ | | | 1.19(1.14,1.25) | $2.39 \times 10^{-13}$ |
| 6 | 41483960 | rs12664490 | | T/C | 0.16 | 0.80(0.75,0.85) | $5.52 \times 10^{-12}$ | 0.81(0.76,0.86) | $1.34 \times 10^{-10}$ | 0.81(0.76,0.86) | $1.24 \times 10^{-10}$ |

All p values are nominal and two-sided.
[a]Data from single-variant analysis in GWAS.
[b]Conditional analysis using GCTA, conditioning on the lead variant in each locus.
[c]Joint analysis using GCTA including the lead variant and the significant variants in conditional analysis.

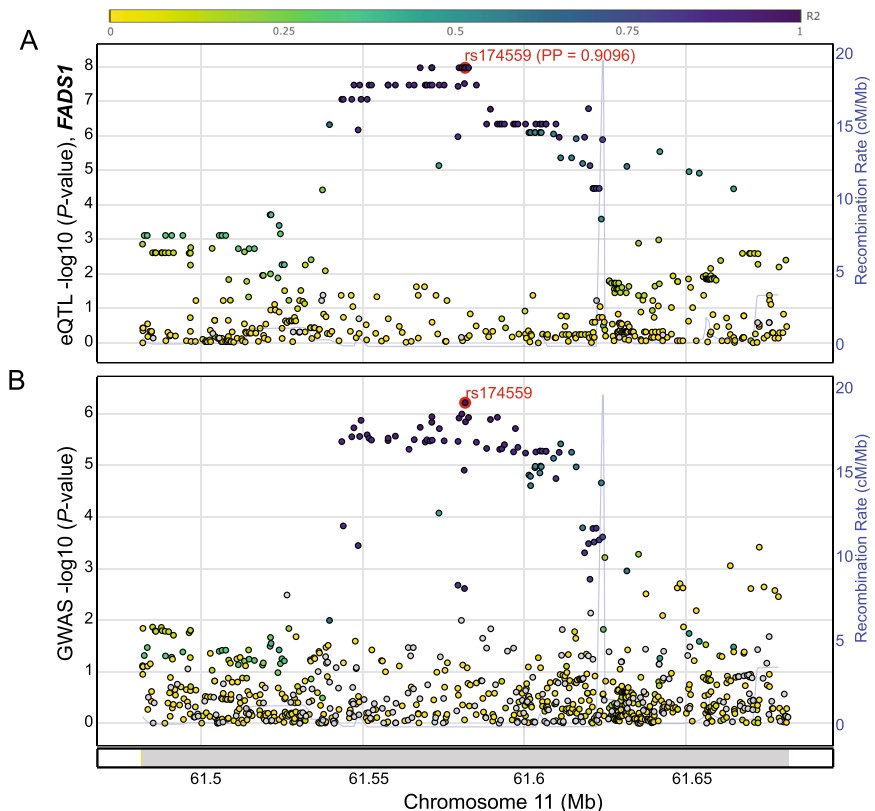

**Fig. 2 | Colocalization of lung adenocarcinoma GWAS signal from the new locus on Chr11 with *FADS1* eQTL signal.** Colocalization analysis was performed using HyPrColoc with summary statistics from Taiwanese lung eQTL data (for *FADS1* gene, **A**) and those of EA GWAS discovery set (**B**). LD R² (1000 Genomes, EA) of each SNP with the GWAS lead SNP, rs174559 (red circle), is color-coded as shown in the top band. Colocalization posterior probability (PP) is shown next to the candidate SNP, rs174559. Note that the p value of rs174559 in GWAS was based on the discovery data and did not include the Japanese replication data. All eQTL p values were two-sided and not adjusted for multiple testing.

which is a key enzyme in the metabolism of polyunsaturated fatty acids and plays a key role in inflammatory diseases[40]. Higher *FADS1* levels in the lung tissues were associated with LUAD risk, which is consistent with its role in increasing the proliferation and migration of laryngeal squamous cell carcinoma through activation of the Akt/mTOR pathway[41]. Among the known loci, colocalization identified *TP63* at 3q28 and *ACVR1B* at 12q13.13 (Supplementary Data 5).

We then performed a TWAS using LCTCNS eQTL dataset. TWAS identified *FADS1* as a susceptibility gene from the 11q12.2 locus (TWAS $P = 3.01 \times 10^{-6}$) validating the finding from the colocalization analysis. We further identified *ELF5* (TWAS $P = 1.89 \times 10^{-8}$) as a novel gene from a locus (at 11p13) not originally passing the genome-wide significance threshold based on a single-variant test in our EA discovery GWAS

(Supplementary data 6, Methods). For these two loci, we also performed TWAS conditional analysis to assess whether genetically predicted expression of these genes explain most of the GWAS signal. When GWAS signal was conditioned on predicted expression of *ELF5*, most of the signal disappeared, adding support for *ELF5* as the main susceptibility gene in this locus (Supplementary Fig. 5A). *ELF5* encodes E74-like factor 5, a key transcription factor of alveologenesis of mammary glands[42]. Lower levels of *ELF5* were associated with LUAD risk in the TWAS. Similarly, when GWAS signal was conditioned on predicted expression of *FADS1*, the strongest part of the signal disappeared (Supplementary Fig. 5B). We further performed TWAS analysis using GTEx lung eQTL dataset (v8, n = 515, ~85% Europeans) and identified five genes from four loci (Supplementary Data 6). While identification

of *ELF5* was common between two datasets, TWAS using GTEx data identified four unique genes from three known loci (*DCBLD1*, *MPZL3*, *JAML*, and *LINC00674*). Notably, *FADS1* was identified only by ancestry-matched LCTCNS eQTL dataset even with a ~ 4 times smaller sample size.

An investigation of the local environment of susceptibility loci revealed further plausible candidate genes that could be pursued in laboratory follow-up. For instance, rs137884934 on 3q22.3 maps to *PIK3CB* encoding an isoform of p110 catalytic subunit of Class IA PI3K[43]. Previous studies have shown that PI3K/Akt/mTOR signaling pathway plays an important role in the development and progression of non-small cell lung cancer[44]. Moreover, rs764014 on 15q21.3 is located adjacent to *NEDD4*, which is a negative regulator of tumor suppressor PTEN[45], which encodes a lipid phosphatase which counteracts the growth promoting effect of PI3K pathway[46].

## Multi-ancestry meta-analysis in East Asian and European populations

To identify variants shared by EA and EUR populations, we performed a fixed effect, multi-ancestry GWAS meta-analysis including data from samples in EA (11,753 cases and 30,562 controls) and samples from EUR populations (11,273 cases and 55,483 controls). We identified four additional loci (Supplementary Table 7) with similar effect sizes in the two populations: rs1130866 (2p11.2, OR = 1.08, $P = 1.56 \times 10^{-8}$), rs2320614 (4q32.2, OR = 1.08, $P = 6.51 \times 10^{-9}$), rs34638657 (16q23.3, OR = 1.09, $P = 2.19 \times 10^{-9}$) and rs638868 (18q12.1, OR = 1.08, $P = 3.6 \times 10^{-8}$). Regional association plots are shown in Supplementary Fig. 6. A multi-ancestry meta-analysis stratified by smoking status did not reveal loci specific to never-smokers or individuals with a history of smoking (sample size information in Supplementary Table 8).

Among the four loci, rs1130866 at 2p11.2 is a missense variant (Ile131Thr) of *SFTPB*, encoding surfactant protein B. Pulmonary surfactant lines the alveoli of lung to reduce the surface tension and is essential for lung function, and increasing circulating level of pro-SFTPB suggested increased lung cancer risk based on prediagnostic samples[47]. Notably, two other novel variants, rs34638657 at 16q23.3 (*MPHOSPH6*)[48,49] and rs2320614 at 4q32.2 (*NAF1*)[50], are on or near genes implicated in telomere biology. Together with other known or new loci (rs2736100 *TERT*, rs4268071 *POT1*, rs75031349 *RTEL1*[51,52], rs7902587 *OBFC1*[53], rs35446936 *TERC*) (Supplementary data 7), our findings further support the role of telomere biology in LUAD.

## Mendelian randomization analysis of telomere length

We performed a Mendelian randomization (MR) analysis to investigate a potential causal relationship between telomere length and the risk of LUAD. The MR analysis was based on 46 independent variants identified in a recent multi-ancestry GWAS of telomere length in the TOPMed study[54], cumulatively accounting for 3.74% of telomere length variance (Methods). Since genetic effects on telomere length showed no evidence of heterogeneity across populations in the TOPMed study, we used the genetic effects estimated based on all populations in the TOPMed study. Our MR analysis was based on MR-PRESSO[55], a robust approach that estimates causal effects after removing variants detected with evidence of pleiotropic effects. Genetically predicted longer telomere length was significantly associated with increased risk of LUAD with similar ORs (per one standard deviation change in genetically increased telomere length) between the two populations: OR = 2.61 (95% CI = 2.08, 3.28, $P = 8.14 \times 10^{-10}$) in EA populations, OR = 2.67 (95% CI = 2.07, 3.43, $P = 7.14 \times 10^{-9}$) in EUR populations, consistent with previous MR reports[56–58] as well as a study of white blood cell DNA telomere length and lung cancer risk in multiple prospective cohorts[59]. MR analyses stratified by smoking status showed similar results between never-smokers and individuals with a history of smoking (Supplementary Table 9). We performed sensitivity analyses using genetic effects estimated based on Asian and European populations in

the TOPMed study separately and found similar results (Supplementary Table 9).

## Comparing the genetics of LUAD in EA and EUR populations

We systematically compared the effect size in EA vs. EUR populations of 38 susceptibility variants for LUAD. These included 12 variants identified in the current study, 26 variants previously reported in EA[10,11,13–15,31] and/or EUR[16,19,20] populations, and results of multi-ancestry meta-analyses combining data from EA and EUR[24] populations (Supplementary Data 8). As expected, ten SNP associations that were independently identified in both populations and through multi-ancestry analysis were very similar (Fig. 3A, B, C). In contrast, out of the 20 SNP associations initially identified in EA populations, two had MAF < 0.01, 11 showed no evidence of association within EUR populations at $P < 0.05$ (Fig. 3D and Fig. 3E, Supplementary Data 8), and 11 associations were significantly different between the two populations with FDR < 0.05. Similar population differences were observed among never-smokers and individuals with a history of smoking (Supplementary Fig. 7). For variants with MAF > 0.01 in both populations, the lack of association in EUR populations did not seem to be driven by low MAF or lower statistical power, as MAFs in both populations for most variants were similar and GWAS in both populations had adequate power to detect at least some evidence of association (Supplementary Data 9). Further, evaluation of gene region plots that spanned 500 kb for these loci within EUR populations showed no or very weak evidence of association for other variants in the region as well as the lead variants from the EA populations (Supplementary Fig. 8A–J), with one exception (Supplementary Fig. 8K). For 8 SNPs initially identified in EUR populations, there was evidence of association for 5 variants in EA populations (Fig. 3F, Supplementary Fig. 9) although all variants were attenuated in the EA compared to the EUR population and one variant had MAF <1% in EA; moreover, two variants were significantly weaker (Supplementary Data 8, Supplementary Fig. 9). Similar patterns were observed among never-smokers and individuals with smoking history (Supplementary Fig. 7).

We used LDSC[27] to evaluate the heritability and genetic correlation between individuals with a history of smoking and never-smokers within each population and POPCORN[60] across populations. The genetic correlation was weaker between never-smokers in EA and EUR populations compared to individuals with a history of smoking (Supplementary Fig. 10) although power was limited given the relatively small sample sizes within each group (Supplementary Table 8). Larger sample sizes are needed to estimate these characteristics more precisely.

## Polygenic risk score and gene-smoking interaction analysis

We investigated whether the polygenic risk score (PRS), which was based on the cumulative effect of 25 independent susceptibility loci for LUAD in EA (Supplementary Table 4), interacted with smoking status to influence the risk of LUAD, given previous evidence of gene-environment interaction[61,62]. Since only summary statistics were available for some datasets (instead of individual genotype data), we developed a statistical method for testing the multiplicative smoking-PRS interaction using the summary statistics for the susceptibility variants (Methods). Compared to the middle quintile that represents the average risk in the general population, the top quintile had OR of 2.07 (95% CI = 1.99, 2.15) for never-smokers and 1.80 (95% CI = 1.70, 1.89) for individuals with a history of smoking ($P_{interaction} = 0.0058$, Fig. 4, Supplementary Fig. 11), providing statistical evidence that the association between PRS and LUAD risk was higher for never-smokers. Moreover, we tested for the presence of multiplicative interactions between smoking status and each individual susceptibility variant in the PRS and found five variants with stronger associations in never-smokers than in individuals with a history of smoking ($P < 0.05$) (Supplementary Table 2).

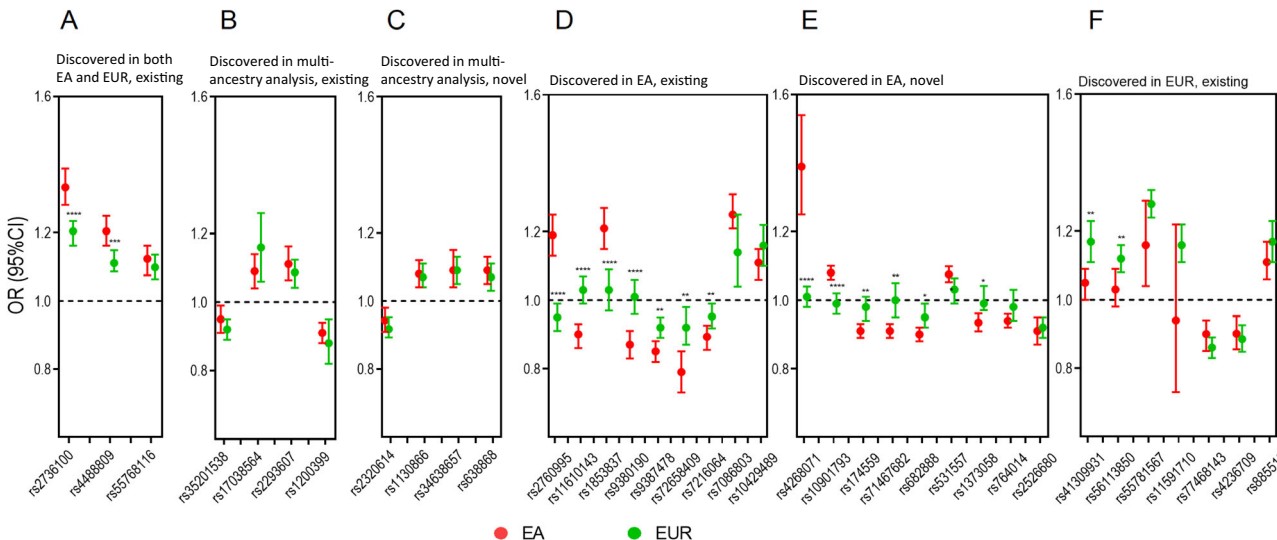

**Fig. 3 | Comparing odds ratios (ORs) of lung adenocarcinoma susceptibility variants between East Asian (EA) and European (EUR) populations.** Here, the effect allele was defined as the minor allele in EA. Each error bar represents the 95% confidence interval of the OR (the center). **A** Susceptibility variants previously discovered (at genome-wide significance) in both EA and EUR populations. **B** Variants previously identified by multiple-ancestry meta-analysis of Chinese and EUR populations; **C** Variants were identified by multiple-ancestry meta-analysis combining EA samples in our study and EUR samples in ILCCO. **D** Variants identified only in EA populations. **E** Novel variants identified in the current study; **F** Variants identified only in EUR populations. Variants are labeled with *, **, *** and **** corresponding to $0.01 \leq p_{het} < 0.05$, $0.001 \leq p_{het} < 0.01$, $0.0001 \leq p_{het} < 0.001$ and $p_{het} < 0.0001$, respectively; here, $p_{het}$ (t-statistic, two-sided) is the $p$ value for testing the heterogeneity of effect sizes between EA and EUR populations. Sample sizes for EUR populations in all panels: 11,273 cases and 55,483 controls. Sample sizes for EA populations: 11,753 cases and 30,562 controls for (**A**, **B**, **C**, **D**, and **F**); 21,658 cases and 150,676 controls for (**E**).

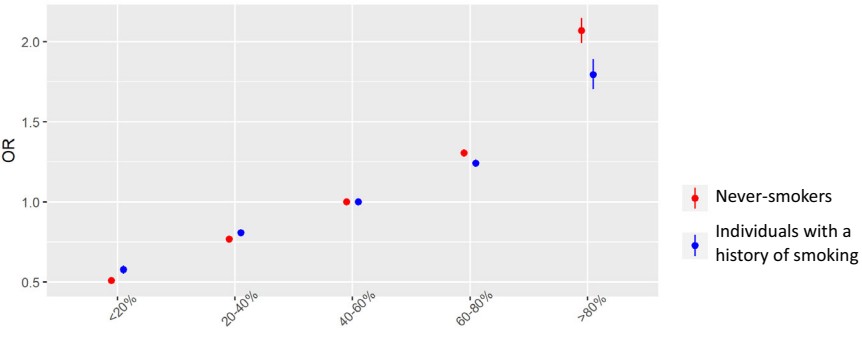

**Fig. 4 | A polygenic risk score (PRS) is more strongly associated with risk of lung adenocarcinoma in never-smokers than in individuals with a history of smoking ($P = 0.0058$).** The PRS was defined based on 25 independent variants that achieved genome-wide significance in EA with weights derived from the meta-analysis of the current study (Supplementary Table 4). The odds ratios (ORs) and the standard errors of the 12 novel variants were based on 21,658 cases and 150,676 controls. The ORs and the standard errors of the other 13 variants were based on 11,753 cases and 30,562 controls. The figure shows the ORs and their 95% confidence intervals comparing each quintile group to the middle quintile for individuals with a history of smoking (blue) and never-smokers (red).

## Genetic architecture, performance of PRS and sample size requirements in EA populations

To further investigate the underlying genetic architecture of susceptibility (Methods) to LUAD[63] in EA populations, we performed a GENESIS[26] analysis based on the GWAS summary statistics for our larger never-smoker dataset. We estimated that ~2275 (s.e. = 1167) susceptibility variants are independently associated with LUAD, suggesting that LUAD is a highly polygenic disease and most of the susceptibility variants have very small effect sizes. Based on the estimated parameters, we investigated how the performance of a PRS, measured as the area under the receiver operating characteristic curve (AUC), depended on the sample size of the training GWAS (Fig. 5). The AUC is predicted to be 60.7% (95% CI = 56.6%, 64.8%) at the current sample size and will increase to 66.9% (95% CI = 62.5%, 71.3%) when the sample size increases to 70,000 cases with one control per case and 68.4%

(95% CI = 64.0%, 72.8%) with 1,000,000 controls. Of note, even a small increase of AUC value for a PRS can help identify many more subjects at risk[64].

## Discussion

We conducted the largest GWAS of LUAD in an EA population to date and identified 12 novel susceptibility variants achieving genome-wide significance. In addition, two variants identified from a previous multi-ancestry meta-analysis achieved genome-wide significance as well in EA alone after we combined the reported summary data with our independent data. In total, including the previously described genetic variants, 28 variants at 25 loci have reached genome-wide significance for LUAD in EA populations, representing major progress in elucidating the genetic basis of LUAD. Finally, a multi-ancestry meta-analysis identified four

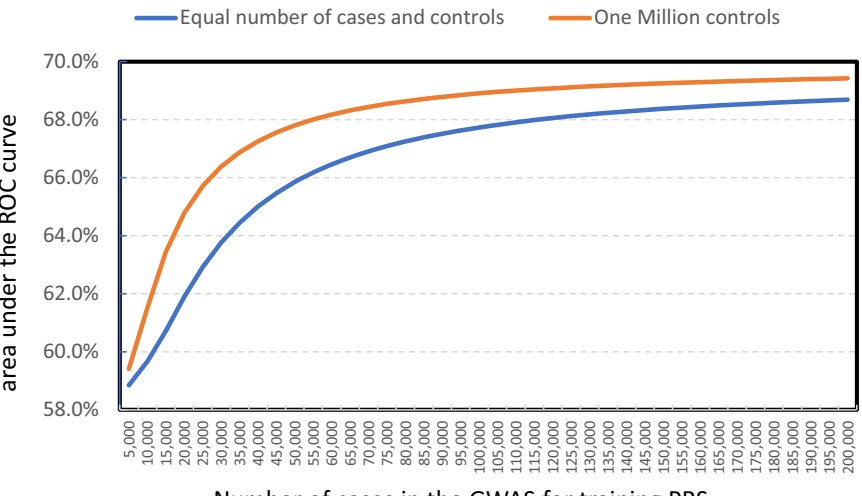

**Fig. 5 | The expected area under the receiver operating characteristic curve (AUC) of a polygenic risk score (PRS) built based on a GWAS of specified sample sizes for lung adenocarcinoma in never-smoking East Asians.** For "1 million controls", the x-coordinate represents the number of cases, assuming the study has 1 million controls. For "Equal number of cases and controls", the x-coordinate represents the numbers of cases, assuming the same number of cases and controls.

additional loci in the combined EA and EUR populations, with consistent effects in both.

Our eQTL colocalization and TWAS analyses using an ancestry-matched lung eQTL dataset (EA population) identified novel LUAD susceptibility genes including *FADS1* and *ELF5*. Importantly, *FADS1* is regulated by sterol-response element-binding proteins (SREBPs)[65], which govern lipid metabolism in alveolar type II (ATII) cells[66]. *ELF5* is also expressed in tissues with glandular/secretory epithelial cells including salivary gland and lung[67,68] and 3.2% of lung alveolar type II cells express *ELF5* in GTEx single-cell expression data. Identification of *FADS1* and *ELF5* in our study suggests a role for alveolar lineage-specific genes and pathways in LUAD susceptibility. Notably, the missense variant (Ile131Thr), rs1130866, in *SFTPB* identified through the multi-ancestry analysis was a protein quantitative trait locus (pQTL) for SFTPB in blood[69], where the LUAD risk-associated A allele (Ile131) is correlated with increased SFTPB levels. Importantly, the genomic region encompassing rs1130866 presents weak LD and high SNP density, consistent with the presence of a recombination hot spot[70], and therefore fine-mapping inspecting low-frequency variants in the region is warranted. Our TWAS analyses using both ancestry-matched and ancestry-discordant lung eQTL datasets identified both common and unique genes from each dataset, highlighting potential benefits of an eQTL dataset of larger sample size and the importance of an ancestry-matched eQTL dataset, even at a smaller sample size, in detecting susceptibility genes.

We evaluated the presence of a gene-environment interaction with tobacco smoking in our EA data. We found that the association between a PRS (constructed by the lead variants at the 25 loci with genome-wide significance in EA) and LUAD in never-smokers was statistically significantly stronger than in individuals with a history of smoking (Fig. 4). This finding, together with our recent paper showing a stronger association of PRS for LUAD risk in non-coal users than in coal users[71], provides evidence that genetic susceptibility may vary by exposure patterns in EA populations.

We systematically compared top GWAS findings that had been initially reported in one or the other or both populations. After accounting for differences in MAFs and statistical power as well as the local LD pattern of each locus (500 kb each side of the lead variant), we found that a substantial number of the associations initially reported in EA populations showed no signal in EUR populations. It might reflect causal variants for these loci not being tagged well in the EUR

populations. This might also suggest important differences between EA and EUR in the genetic architecture of LUAD samples, which could be caused by differential environmental exposures. Finally, this observation is also consistent with distinct tumor molecular characteristics (e.g., *EGFR* mutation prevalence was higher in Asians than EUR populations) observed in LUAD suggesting different etiologies influenced by genetic and/or environmental factors[13,72,73].

Our genetic architecture analysis suggested that LUAD is a highly polygenic disease. Expanding GWAS of LUAD will continue to identify many risk variants albeit with smaller effect sizes. Moreover, our analysis predicts that the AUC of PRS for EA never-smokers could be improved to 66.9% for a GWAS training dataset with 70,000 cases and 70,000 controls that could be further increased with a greater number of controls. Thus, an expanded GWAS in the future can lead to the substantial improvement in knowledge about the underlying genetic architecture of LUAD; increased understanding of how known or suspected lung cancer environmental risk factors interact with genetic susceptibility; and assessment of the potential clinical utility of risk models integrating both genetic and non-genetic risk factors[74,75].

There are several limitations in the current study. First, the discovery phase included subjects of diverse EA populations (Mainland China 38.2%, Japan 45.9%) and the replication phase only included subjects from Japan. However, our data did not show evidence of heterogeneity in effect sizes for susceptibility variants between Han Chinese and Japanese populations or across geographic locations (Supplementary Table 5), suggesting a minimal impact for using a single EA population for replication. Second, we were underpowered to conduct formal heritability correlation analyses to compare the genetic architecture in EA and EUR populations stratified by smoking status; larger studies will be needed to conclusively characterize differences. Furthermore, completely elucidating the genetic basis of ancestry differences requires detailed information about age of onset, family history and exposures. Finally, rs4268071 (Table 2) achieved genome-wide significance in the discovery data but replication data were not available. While the significance was primarily driven by Japanese samples (MAF = 0.04 in Japanese and <1% in other populations), there was no evidence of heterogeneity in effect estimates across EA populations. Replication is warranted to further establish its etiological role.

In conclusion, we identified 12 novel variants in a GWAS of LUAD in EA populations as well as 4 novel variants in a multi-ancestry meta-

analysis of EA and EUR populations. Colocalization and TWAS analyses using an ancestry-matched lung tissue eQTL dataset identified candidate susceptibility genes with suggested roles in alveolar lineage. At the same time, a large majority of variants identified in the EA GWAS showed no evidence of association in EUR populations. Larger samples sizes with data on environmental risk factors will be needed to further characterize the etiologic differences between these populations. Finally, our genetic architecture analysis suggests that the performance and the clinical utility of the PRS will be substantially improved by larger GWAS in the future.

## Methods

### Ethics statement

All participants provided informed consent according to protocols that were evaluated and approved by the internal review boards of the contributing centers. Protocols used to generate new, unpublished data presented in this paper were approved by the National Cancer Center Institutional Review Board, Japan and the Aichi Cancer Center Ethics Committee, Japan.

### Overview of study

We conducted a two-phase GWAS meta-analysis of LUAD in EA populations, including Female Lung Cancer Consortium in Asia (FLCCA), Nanjing Lung Cancer Study (NJLCS)[10,24], National Cancer Center of Japan (NCC) Research Institute and Aichi Cancer Center (ACC). For the FLCCA study, details of the study design, participating studies, case ascertainment, genotyping, and quality controls have been described in detail[9]. Briefly, this international consortium is composed of Asian women who never smoked and resided in Mainland China, Hong Kong, Singapore, Taiwan, South Korea and Japan at the time of recruitment. All were genotyped using the Illumina 660 W, 370 K and 610Q microarrays.

The NCC study included lung cancer patients from NCC and BioBank Japan (BBJ) and non-cancer controls from the Japan Public Health Center-based Prospective Study and the Japan Multi-Institutional Collaborative Cohort Study, genotyped by Illumina HumanOmniExpress and HumanOmni1-Quad genotyping platforms. The ACC study included lung cancer patients from the Aichi Cancer Center, Kyoto University, Okayama University and Hyogo College of Medicine and non-cancer controls from the Nagahama Study and the Aichi Cancer center. Samples were genotyped by Illumina 610k and Illumina660k platforms[15,76]. The NJLCS study at the Nanjing Medical University was based on meta-analysis of three studies: the Nanjing GWAS with subjects from Nanjing and Shanghai, the Beijing study with subjects from Beijing and Wuhan (genotyped by Affymetrix Genome-Wide Human SNP Array 6.0) and the Oncoarray GWAS[10,77,78].

The replication study included cases from multiple sources (BBJ, NCC, Kanagawa Cancer Center, Akita University Hospital, Tokyo Medical and Dental University, Hospital and Gunma University Hospital, and Fukushima Medical University School of Medicine) and non-cancer controls from BioBank Japan. Cases were genotyped using the Invader assay and the control samples in BioBank Japan were genotyped using the Illumina HumanOmniExpress genotyping platform.

For the multi-ancestry meta-analyses of LUAD and cross-population comparison of top GWAS findings with both never-smokers and individuals with a history of smoking, we used 11,273 cases and 55,483 controls of European ancestry in the Integrative Analysis of Lung Cancer Etiology and Risk team of the International Lung Cancer Consortium (INTEGRAL-ILCCO)[16] (Supplementary Table 8). For the multi-ancestry analysis and cross-population comparisons of smokers, we used European samples genotyped with the OncoArray platform in the ILCCO study (Supplementary Table 8). For the multi-ancestry and cross-population comparisons analysis of never-smokers, we used the GWAS of European never-smoking subjects from Hung et al.[21].

### Quality control, imputation and association analysis in EA populations

For each study, SNPs with minor allele frequency (MAF) < 0.01, Hardy-Weinberg Equilibrium (HWE) $p$ value < $10^{-6}$ in controls were removed; subjects with missing rate >3%, sex discrepancy, or displaying non-East Asian ancestry based on principal component analysis scores were removed. Moreover, for any pairs of subjects estimated to be related with identity by descent pihat >0.10 using PLINK (V2.0), we removed one subject. Imputation was performed using IMPUTE2 and the 1000 Genomes Project East Asian samples (Phase 3) as reference. After imputation, SNPs with imputation quality score ≥ 0.5 were used for association analysis in each study. Logistic regression under an additive model was performed using SNPTest (V2) or PLINK2 based on imputed genotypic dosage data adjusting for smoking (if both smokers and never smokers were present) and PCA scores to control for population stratification. Meta-analysis was performed using inverse-variance weighted fixed effects methods. All $p$ values were two-sided. We consider the following variants as novel for the GWAS in EA: (1) the lead variant with $p < 5 \times 10^{-8}$ in a locus that has not been previously reported in either EA or EUR populations, or (2) a secondary variant with $p < 5 \times 10^{-8}$ conditioning on the lead variant in a previously reported locus in either EA or EUR populations with the requirement that the LD $R^2 \leq 0.2$ between the secondary and the lead variants in both populations.

LDSC[27] was used to estimate the heritability attributed to genome-wide common variants and to assess the potential inflation due to insufficient correction of population stratification. LDSC was also used to estimate the genetic correlation of LUAD between never-smokers and individuals with a history of smoking in each population. We used POPCORN[60] to estimate the genetic correlation between EA and EUR populations because LD patterns are expected to be different. To account for the difference of allele frequencies in the two populations, we also used POPCORN to estimate the cross-population genetic-impact correlation that was defined as the correlation of population specific phenotypic variance explained by each SNP.

### Conditional analysis and fine mapping

To identify independently associated SNPs at an established susceptibility locus, we performed conditional analysis using software Genome-wide Complex Trait Analysis (GCTA)[79] based on the GWAS meta-analysis summary results of EA populations. LD for the conditional analysis was calculated using a reference population of 4544 controls from the FLCCA study to achieve a desirable accuracy. Here, genotypes for FLCCA were imputed using IMPUTE2 and the 1000 Genomes Project (Phase 3) reference samples with EA ancestry. SNPs with imputation quality <0.5 were excluded from the reference set for conditional analysis. Conditional analysis was restricted to 14 loci with lead SNPs achieving genome-wide significance in the discovery-phase meta-analysis. We did not perform conditional analyses for other new SNPs that did not achieve genome-wide significance in the discovery-phase meta-analysis because secondary SNPs would not survive multiple testing correction. Conditional analysis was restricted to SNPs less than 500 kb from the lead SNP of each locus. To identify multiple potentially independent SNPs in one locus, we performed stepwise conditional analysis using GCTA. All SNPs identified with $P < 5 \times 10^{-8}$ and the lead SNP of the locus were put into one model to derive the joint estimate of ORs, appropriately adjusting for LD among all SNPs. Only SNPs with $p$ value < $5 \times 10^{-8}$ in both conditional and joint analyses were considered to be independently associated SNPs.

For 11 out of the 14 loci with genome-wide significance in the discovery phase, we performed a Bayesian fine-mapping analysis using FINEMAP[32] to nominate 95% credible set variants using the same set of imputed genotypes of 4544 FLCCA control subjects as an LD reference. We did not perform fine-mapping analysis for two loci in MHC regions, because of the complex and extensive LD patterns in this region. We

also excluded the locus at 7q31 because the lead SNP, rs4268071, had MAF < 1% in our LD reference population. MAF of this variant is 4% in the Japanese populations (45.8% of cases and 74.5% of controls in the discovery set) but <1% in other EA populations included in our study. For FINEMAP analysis, we tested the variants within ±500 kb of the lead SNP and set the number of maximum causal variants as the number of independent signals ($P \leq 10^{-5}$) observed in the conditional analysis for each locus.

## Proportion of familial risk explained

We considered a set of identified variants for LUAD. For SNP t, we defined $p_t$ as the frequency of the risk allele and $OR_t$ as the estimated per-allele odds ratio. Under a multiplicative model, the fraction of the familial risk explained by the set of SNPs was calculated as $\sum_t \log(\lambda_t) / \log(\lambda_0)$, where $\lambda_0$ is the observed familial risk to the first degree of LUAD cases and $\lambda_t$ is the familial risk due to the $t^{th}$ SNP:

$$\lambda_t = \frac{p_t OR_t^2 + (1 - p_t)}{(p_t OR_t + 1 - p_t)^2}. \tag{1}$$

## Heritability partitioning in functional classes and tissue-specific analyses

Stratified LD score regression (sLDSC)[80] was conducted to identify functional annotations enriched for LUAD heritability using summary statistics from the discovery phase of meta-analysis in EA populations. In addition to the functional annotations provided by the sLDSC package, we also analyzed the gene sets defined by smoking studies: differentially expressed genes in peripheral blood mononuclear cells upon nicotine treatment ("PBMC nicotine" gene set) from Moyerbrailean et al.[81], those in non-tumorous lungs between current- and never-smokers ("Lung smoking" gene set) from Bosse et al.[82], and those in normal bronchial airway epithelial cells between current- and never-smokers ("Airway smoking" gene set) from Beane et al.[83]. An annotation was considered to be significantly enriched for LUAD heritability if FDR < 0.05.

We then performed sLDSC to prioritize relevant tissue types (lung, blood/immune, and brain/CNS) using tissue-specific expressed genes from GTEx v6p (53 tissue types) and other public expression datasets (152 tissue types), as well as tissue-specific chromatin annotations from EnTEX (111 annotations in 26 tissue types) and Roadmap dataset (378 annotations in 85 tissue types) as described by Finucane and colleagues[36]. We used GTEx v6p expression data based on a comparison with v8 data, where a median of 83% of tissue-specific differentially expressed genes were shared between two versions. In general, we did not find significant enrichment for individual annotations after adjusting for the multiple testing. To increase the power of prioritizing relevant tissues (lung, blood/immune, and brain/CNS), we performed an aggregated analysis to test if $p$ values from one tissue (e.g., lung) tended to be smaller than those from the other two tissue groups (blood/immune, and brain/CNS) using the Wilcoxon rank test.

## eQTL colocalization analysis and TWAS

EA lung eQTL dataset is based on a cohort of 115 never-smoking LUAD patients from Taiwan, referred to as LCTCNS (Lung cancer tissue cohort of never-smokers). Expression array data was obtained for non-tumor lung tissues of these patients using the Illumina WG-DASL HumanRef-8 v3 or HumanHT-12 v4 BeadChip (Illumina Inc.) (Gene Expression Omnibus accession number GSE46539)[84]. Genotype data from buffy coat DNA was obtained using the Illumina Human660W-Quad BeadChip. A systematic quality control for the genotype data was performed as previously described[12] (SNPs were excluded if call rate <90%, MAF < 5%, or $P < 0.0001$ based on the Hardy-Weinberg equilibrium test. Samples were excluded if call rate <90%, sex discrepancies based on the X chromosome heterozygosity, contaminated samples

with high heterozygosity scores, or first or second- degree relatives), and imputation was carried out using Minimac4 (V4.0.3) with the 1000 Genomes reference set (all populations). For eQTL analysis, expression data was processed for background correction as previously described[84]. Briefly, we kept the probes that are present in both the BeadChip platforms and further removed those with low expression levels (detection $p > 0.05$). Based on the data at the remaining 24,216 probes, we applied model-based background correction. Log$_2$-transformed expression levels of 24,216 probes were then used to obtain 20 latent factors based on probabilistic estimation of expression residuals (PEER) while specifying batch, sex, age, medical operation status, RNA integrity number, and RNA input quantity as known confounders. The expression residuals from PEER were then inverse rank transformed to the standard normal distribution (the inverse rank transformed residuals) and were used as the dependent variable in the expression levels for eQTL analysis. eQTL analysis was conducted for 29 GWAS lead SNPs (all EA loci including discovery, replication, and conditional signals plus new loci from the multi-ancestry GWAS). In LCTCNS, all these SNPs have a MAF of >0.01. For each GWAS lead SNP, its association with each probe located within ±500 kb of the SNP was tested using an additive linear model where the dependent variable was the expression level as described above and the independent variable was the effect allele count. Based on the resulting $p$ values of these eQTL analyses for all 29 SNPs, the corresponding Benjamini–Hochberg FDR was calculated. Colocalization analysis was performed using eCAVIAR[37] and HyPrColoc[38] via ezQTL platform for eight GWAS lead SNP-eQTL gene pairs displaying FDR < 0.05 in LCTCNS (Supplementary Data 5). For each of these eight SNP-probe pairs, we further examined the association between the probe and SNPs within ±100 kb of the lead SNP using Matrix eQTL to obtain the summary statistics as an input to ezQTL for colocalization analysis using HyPrColoc and eCAVIAR. For loci on MHC regions, ±10 kb window was used for computational efficiency of colocalization analyses. LD matrix was obtained from 1000 Genomes EA populations. For HyPrColoc, posterior probability of >0.7 was used as a cutoff for colocalization. For eCAVIAR analysis, colocalization posterior probability (CLPP) score > 0.01 was used as a cutoff for colocalization.

For TWAS, we adopted FUSION[85] using LCTCNS or GTEx v8 lung eQTL data and summary statistics of EA discovery GWAS meta-analysis. We computed weights using the elastic-net regression model for 24,216 expression probes (LCTCNS) or 24,687 genes (GTEx v8 lung) and cis-SNPs within 500 kb of the gene for each probe. LD matrix was obtained from 1000 Genomes EA populations. We performed association analysis for 1875 expression probes (LCTCNS) or 5534 genes (GTEx v8 lung) with cross-validation cutoff of $R^2 > 0.05$ based on the elastic-net model. We defined a significant transcriptome-wide association as TWAS $P < 2.6 \times 10^{-5}$ (0.05/1875; LCTCNS) or $P < 9 \times 10^{-6}$ (0.05/5534; GTEx v8 lung) based on Bonferroni correction. For two loci passing this cutoff from LCTCNS analysis (*ELF5* and *FADS1*), we further performed conditional analysis as implemented in FUSION by conditioning the GWAS signal on the predicted expression of the probe with the best TWAS $P$ value.

## Mendelian randomization

We performed MR analysis to investigate the potential causal relationship between telomere length and the risk of LUAD. MR analysis was based on 46 common SNPs identified in a recent multi-ancestry meta-analysis of telomere length in the TOPMed[54] study. The original paper identified 48 variants associated with telomere length that collectively explained 4.35% of telomere length variance; two of them at the *TERT* locus were excluded using the LD filter $R^2 < 0.05$ that together explained 0.61% of the telomere length variance; the remaining 46 variants included in our MR analysis explained 3.74% of telomere length variance. Because there was no significant heterogeneity of effect sizes on telomere length across populations (Table S4 in

Taub et al.[54]), the primary MR analyses were based on the estimated effect sizes combining all samples in the TOPMed study in a joint regression model for telomere length. Analyses were based on MR-PRESSO[86], a powerful and robust approach designed to deal with widespread horizontal pleiotropy. This approach uses a formal testing framework to (1) detect the presence of horizontal pleiotropy, (2) detect variant outliers, (3) evaluate distortion, and (4) re-estimate causal effect sizes after removing potentially problematic variants. According to simulations, this approach is best suited when horizontal pleiotropy occurs in <50% of instruments. This approach identified 5–7 outlier variants in our data. The estimated $\beta$ from MR analysis was converted as OR, interpreted as risk increase per standard deviation (640 base pairs[87]) increase of the genetically predicted telomere length.

## Testing the interaction between polygenic risk score and smoking status

We investigated whether the PRS, which was calculated based on 25 independent SNPs associated with LUAD in EA populations (Supplementary Table 4, excluding three variants identified by conditional analysis), interacted with smoking status for LUAD risk. Because we have only GWAS summary statistics instead of individual-level data for smokers and never-smokers, we developed a statistical method for testing the interaction using summary statistics separately from smokers and never-smokers. Suppose that we have $n^{1+}$ smoking cases, $n^{0+}$ never-smoking cases, $n^{1-}$ smoking controls and $n^{0-}$ never-smoking controls. Let $x_{it}^{s+}$ and $x_{jt}^{s-}$ be the genotype of SNP $t$ for the $i^{th}$ case and the $j^{th}$ control, where $s = 1$ indicates smokers and 0 indicates never-smokers. Given smoking status $s$, we define $PRS_i^{s+} = \sum_{t=1}^{T}\beta_t x_{it}^{s+}$ and $PRS_j^{s-} = \sum_{t=1}^{T}\beta_t x_{jt}^{s-}$ as the PRS for cases and controls, respectively. For smokers ($s = 1$), the association between PRS and disease risk can be quantified as:

$$\Delta_1 = \frac{1}{n^{1+}}\sum_{i=1}^{n^{1+}}PRS_i^{1+} - \frac{1}{n^{1-}}\sum_{j=1}^{n^{1-}}PRS_j^{1-}, \quad (2)$$

the difference of average PRS between cases and controls. Similarly, we define $\Delta_0$ to be the difference of average PRS between cases and controls for never-smokers. Testing the PRS*smoking interaction can be done using $Z = \frac{\Delta_1 - \Delta_0}{\sqrt{var(\Delta_1^2) + var(\Delta_0^2)}}$. Under the null hypothesis of no interaction for all variants, $Z \sim N(0,1)$ asymptotically. Assuming SNPs are independent, we derive $Z = \sum_{t=1}^{T}(w_t^1 z_t^1 - w_t^0 z_t^0)$, where $z_t^s$ is the z-score for testing association for SNP $t$ in subjects with smoking status $s$. The weight is given as

$$w_t^s = \frac{\beta_t \sqrt{\frac{(\sigma_t^{s+})^2}{n_+^s} + \frac{(\sigma_t^{s-})^2}{n_-^s}}}{\sqrt{\sum_{t=1}^{T}\beta_t^2 \left(\frac{(\sigma_t^{1+})^2}{n_+^1} + \frac{(\sigma_t^{1-})^2}{n_-^1} + \frac{(\sigma_t^{0+})^2}{n_+^0} + \frac{(\sigma_t^{0-})^2}{n_-^0}\right)}}. \quad (3)$$

Here, $(\sigma_t^{s+})^2$ and $(\sigma_t^{s-})^2$ are the genotypic variances for SNP $t$ in cases and controls, respectively.

We note that both discovery and replication data are included for testing PRS smoking interaction novel variants included in our PRS to maximize the power of statistical testing. In particular, only the discovery data were available and included for previously identified variants; both discovery and replication data were included for new variants to increase the statistical power. To do this, $w_t^s$ was modified to have SNP-specific sample sizes. All analyses were done using R (x64 4.1.0).

## GENESIS analysis for projecting yield of future expanded studies

The genetic architecture of a disease is defined as the number of susceptibility SNPs and the distribution of their effect sizes[26]. When these parameters are estimated, one can estimate the number of variants achieving genome-wide significance and the accuracy of a polygenic risk model trained using a GWAS with a given sample size. In the current study, we estimated the genetic architecture using GENESIS (GENetic EStimation and Inference in Structured samples)[26] based on the GWAS summary statistics with LD scores calculated based on the genotypes of the subjects of EA ancestry in the 1000 Genomes Project. Since GENESIS requires a large sample size to derive reliable estimates, we performed analysis only for never-smokers in EA. The three-component model $\beta_m \sim \pi p_1 N(0, \sigma_1^2) + \pi p_2 N(0, \sigma_2^2) + (1 - \pi)\delta_0$ best fit the never-smoker data in EA, where $\beta_m$ represents effects sizes, $\pi$ denotes the fraction of truly associated variants in the genome, $\delta_0$ denotes the point mass at zero, $\sigma_i^2$ denotes the variance of effect sizes for the $i^{th}$ component, $\pi p_i$ ($i = 1,2$) represents the fraction of variants with effect size following $N(0, \sigma_i^2)$. Based on this estimated genetic architecture, we calculated the expected number of variants reaching genome-wide significance for a given GWAS and calculated the expected area under the receiver operating characteristic curve (AUC) for an additive polygenic risk prediction model built based on a discovery GWAS for a given sample size. The uncertainty of the AUC was induced by the uncertainty in the estimated parameters in GENESIS ($\Gamma = (\pi, p_1, p_2, \sigma_1^2, \sigma_2^2)$) because of the limited sample size in our summary data. We used a resampling approach to estimate the standard error of AUC. Briefly, we randomly simulated 1000 sets of parameters $\Gamma^k$ given the estimated $\hat{\Gamma}$ and the estimated covariance matrix, and calculated $AUC_k$ for each simulated parameter $\Gamma^k$ for a given sample size. The standard error was calculated based on the 1000 sets of AUC values.

## Reporting summary

Further information on research design is available in the Nature Portfolio Reporting Summary linked to this article.

## Data availability

All data supporting the findings described in this paper are available in the paper and in the Supplementary Information and from the corresponding author or as otherwise indicated upon request. Full TWAS results are included in Supplementary Data 6. The summary statistics for the meta-analysis of the 4 GWAS datasets in East Asian populations for SNPs with $p \leq 0.01$ are in Supplementary Data 10. The results of the replication study for the 38 SNPs tested and the meta-analysis with the GWAS data are in Supplementary Data 11. For the FLCCA study, the GWAS summary data for SNPs with $p < 0.01$ in the study and all SNPs with genome-wide significance in the meta-analysis of East Asian samples are in Supplementary Data 12. The individual genotype data for the FLCCA data are in dbGaP phs000716.v1.p1 (Genome-Wide Association Study of Lung Cancer Susceptibility in Never-Smoking Women in Asia). For the NJLCS study, the GWAS summary data for SNPs with $p < 0.01$ in the study and all SNPs with genome-wide significance in the meta-analysis of East Asian samples are in Supplementary Data 13. For the NCC and ACC studies, please contact Kouya Shiraishi at kshirais@ncc.go.jp or Takashi Kohno at tkkohno@ncc.go.jp for summary statistics. The GWAS data for the European populations contributing to this study are available at dbGap under accession phs000877.v1.p1 (Transdisciplinary Research Into Cancer of the Lung (TRICL), https://www.ncbi.nlm.nih.gov/projects/gap/cgi-bin/study.cgi?study_id=phs000876.v2.p1), phs001273.v3.p2 (Oncoarray Consortium, https://www.ncbi.nlm.nih.gov/projects/gap/cgi-bin/study.cgi?study_id=phs001273.v3.p2). To gain access to all data in dbGaP cited in this paper, please apply for dbGaP Authorized Access. The expression data of the lung cancer tissue cohort of never-smokers in Taiwan are publicly available at Gene Expression Omnibus under accession number GSE46539. The expression and eQTL data from GTEx (v6 and v8) are available from https://gtexportal.org/home/datasets.

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

## Acknowledgements

This work utilized the computational resources of the NIH HPC Biowulf cluster. (http://hpc.nih.gov).

Female Lung Cancer Consortium in Asia (NCI): This study was supported by a Grant-in-Aid for Scientific Research on Priority Areas from the Ministry of Education, Science, Sports, Culture and Technology of Japan, a Grant-in- Aid for the Third Term Comprehensive 10-Year Strategy for Cancer Control from the Ministry Health, Labor and Welfare of Japan, by Health and Labor Sciences Research Grants for Research on Applying Health Technology from the Ministry of Health, Labor and Welfare of Japan, by the National Cancer Center Research and Development Fund, the National Research Foundation of Korea (NRF) grant funded by the Korea government (MEST) (grant No. 2011-0016106), a grant of the National Project for Personalized Genomic Medicine, Ministry for Health & Welfare, Republic of Korea (A111218-11-GM04), the Program for Changjiang Scholars and Innovative Research Team in University in China (IRT_14R40 to K.C.), the National Science & Technology Pillar Program (2011BAI09B00), MOE 111 Project (B13016), the National Natural Science Foundation of China (No. 30772531, and 81272618), Guangdong Provincial Key Laboratory of Lung Cancer Translational Medicine (No. 2012A061400006), Special Fund for Research in the Public Interest from the National Health and Family Planning Commission of PRC (No. 201402031), and the Ministry of Science and Technology, Taiwan (MOST 103-2325-B-400-023 & 104-2325-B-400-012). The Japan Lung Cancer Study (JLCS) was supported in part by the Practical Research for Innovative Cancer Control from Japan Agency for Medical Research and Development (15ck0106096h0002) and the Management Expenses Grants from the Government to the National Cancer Center (26-A-1) for Biobank. BioBank Japan was supported by the Ministry of Education, Culture, Sports, Sciences and Technology of the Japanese government. The Japan Public Health Center-based prospective Study (the JPHC Study) was supported by the National Cancer Center Research and Development Fund (23-A- 31[toku], 26-A-2, 29-A-4, and 2020-J-4) (since 2011) and a Grant-in-Aid for Cancer Research from the Ministry of Health, Labour and Welfare of Japan (from 1989 to 2010). The Taiwan GELAC Study (Genetic Epidemiological Study for Lung AdenoCarcinoma) was supported by grants from the National Research Program on Genomic Medicine in Taiwan (DOH99-TD-G-111-028), the National Research Program for Biopharmaceuticals in Taiwan (MOHW 103-TDUPB-211-144003, MOST 103-2325-B-400-023) and the Bioinformatics Core Facility for Translational Medicine and Biotechnology Development (MOST 104-2319-B-400-002). This work was also supported by the Jinan Science Research Project Foundation (201102051), the National Key Scientific and Technological Project (2011ZX09307-001-04), the National Natural Science Foundation of China (No.81272293), the State Key Program of National Natural Science of China (81230067), the National Research Foundation of Korea (NRF) grant funded by the Korea government (MSIP) (No. NRF- 2014R1A2A2A05003665), Sookmyung Women's University Research Grants, Korea (1-1603-2048), Agency for Science, Technology and Research (A*STAR), Singapore and the US National Institute of Health Grant (1U19CA148127-01). The overall GWAS project was supported by the intramural program of the US National Institutes of Health/National Cancer Institute. The following is a list of grants by study center: SKLCS (Y.T.K.)—National Research Foundation of Korea (NRF) grant funded by the Korea government (MEST) (2011-0016106). (J.C.) – This work was supported by a grant from the National R&D Program for Cancer Control, Ministry of Health &Welfare, Republic of Korea (grant no. 0720550-2). (J.S.S)—grant number is A010250. WLCS (T.W.)—National Key Basic Research and Development Program (2011CB503800). SLCS (B.Z.)—National Nature Science Foundation of China (81102194). Liaoning Provincial Department of Education (LS2010168). China Medical Board (00726). GDS (Y.L.W.)—Foundation of Guangdong Science and Technology Department (2006B60101010, 2007A032000002, 2011A030400010). Guangzhou Science and Information Technology Bureau (2011Y2-00014). Chinese Lung Cancer Research Foundation, National Natural Science Foundation of China (81101549). Natural Science Foundation of Guangdong Province (S2011010000792). TLCS (K.C., B.Q.)—Program for Changjiang Scholars and Innovative Research Team in University (PCSIRT), China (IRT1076). Tianjin Cancer Institute and Hospital. National Foundation for Cancer Research (US). FLCS (J.C.W., D.R., L.J.)—Ministry of Health (201002007). Ministry of Science and Technology (2011BAI09B00). National S&T Major Special Project (2011ZX09102-010-01). China National High-Tech Research and Development Program (2012AA02A517, 2012AA02A518). National Science Foundation of China (30890034). National Basic Research Program (2012CB944600). Scientific and Technological Support Plans from Jiangsu Province (BE2010715). NLCS (H.S.)—China National High-Tech Research and Development Program Grant (2009AA022705). Priority Academic Program Development of Jiangsu Higher Education Institution. National Key Basic Research Program Grant (2011CB503805). GEL-S (A.S.)—National Medical Research Council Singapore grant (NMRC/0897/2004, NMRC/1075/2006). (J.Liu)—Agency for Science, Technology and Research (A*STAR) of Singapore. GELAC (C.A.H.)—National Research Program on Genomic Medicine in Taiwan (DOH98-TDG-111-015). National Research Program for Biopharmaceuticals in Taiwan (DOH 100- TD-PB-111-TM013). National Science Council, Taiwan (NSC 100- 2319-B-400-001). YLCS (Q.L.)—Supported by the intramural pro- gram of U.S. National Institutes of Health, National Cancer Institute. SWHS (W.Z., W-H.C., N.R.)—The work was supported by a grant from the National Institutes of Health (R37 CA70867, UM1 CA182910) and the National Cancer Institute intramural research program, including NCI Intramural Research Program contract (N02 CP1101066). JLCS (K.M., T.K.)—Grants-in-Aid from the Ministry of Health, Labor, and Welfare for Research on Applying Health Technology and for the 3rd-term Comprehensive 10-year Strategy for Cancer Control; by the National Cancer Center Research and Development Fund; by Grant-in-Aid for Scientific Research on Priority Areas and on Innovative Area from the Ministry of Education, Science, Sports, Culture and—Technology of Japan. (W.P.)—NCI R01-CA121210. HKS (J.W.)—General Research Fund of Research Grant Council, Hong Kong (781511 M). The Environment and Genetics in Lung Cancer Etiology (EAGLE), Prostate, Lung, Colon, Ovary Screening Trial (PLCO), and Alpha-Tocopherol, Beta-Carotene Cancer Prevention (ATBC) studies were supported by the Intramural Research Program of the National Institutes of Health, National Cancer Institute (NCI), Division of Cancer Epidemiology and Genetics. ATBC was also supported by U.S. Public Health Service contracts (N01-CN-45165, N01-RC-45035, and N01-RC-37004) from the NCI. PLCO was also supported by individual contracts from the NCI to the University of Colorado Denver (NO1-CN-25514), Georgetown University (NO1-CN-25522), the Pacific Health Research Institute (NO1-CN-25515), the Henry Ford Health System (NO1-CN-25512), the University of Minnesota, (NO1-CN- 25513), Washington University (NO1-CN-25516), the University of Pittsburgh (NO1-CN-25511), the University of Utah (NO1-CN- 25524), the Marshfield Clinic Research Foundation (NO1-CN- 25518), the University of Alabama at Birmingham (NO1-CN- 75022), Westat, Inc. (NO1-CN-25476), and the University of California, Los Angeles (NO1-CN-25404). The Carotene and Retinol Efficacy Trial (CARET) is funded by the National Cancer Institute, National Institutes of Health through grants U01-CA063673, UM1-CA167462, and U01-CA167462. The Cancer Prevention Study-II (CPS-II) Nutrition Cohort was supported by the American Cancer Society. The NIH Genes, Environment and Health Initiative (GEI) partly funded DNA extraction and statis- tical analyses (HG-06-033-NCI-01 and RO1HL091172-01), genotyping at the Johns Hopkins University Center for Inherited Disease Research. This research was supported by the National Research Foundation of Korea (NRF) grant funded by the Korea

government (MSIT) (No. 2020R1A2C4002236). Genotyping of the samples in NJLCS was supported by the National Natural Science of China (81820108028).

Female Lung Cancer Consortium in Asia (Tianjin): Tianjin Science and Technology Committee Foundation, 18YFZCSY00520.

Female Lung Cancer Consortium in Asia (Taiwan): The Ministry of Health and Welfare grants DOH97-TD-G-111-028 (I.S.C.), DOH98-TD-G-111-017 (I.S.C.), DOH99-TD-G-111-014 (I.S.C.); DOH97-TD-G-111-026 (C.A.H.), DOH98-TD-G-111-015 (C.A.H.), DOH99-TD-G-111-028 (C.A.H.); National Health Research Institutes grants NHRI-PH-110-GP-01, NHRI-PH-110-GP-03; and the Ministry of Science and Technology grants MOST108-2314-B-400-038(C.A.H.), MOST109-2740-B-400-002(C.A.H.).

The GWAS of lung cancer in European never smokers was supported by NIH R01 CA149462 (O.Y.G.).

OncoArray study in Europeans: The OncoArray data and analysis from INTEGRAL-ILCCO were supported by NIH U19 CA203654, and U19 CA148127. The data harmonization for ILCCO was supported by Canadian Institute for Health Research (CIHR) Canada Research Chair to R.J.H, and CIHR FDN 167273).

European never-smoking lung cancer study: C.I.A. is a Research Scholar of the Cancer Prevention Institute of Texas (CPRIT) and supported by CPRIT grant RR170048.

Taiwan eQTL study: This study was supported by the Ministry of Health and Welfare grants DOH97-TD-G-111-028 (I.S.C.), DOH98-TD-G-111-017 (I.S.C.), DOH99-TD-G-111-014 (I.S.C.); DOH97-TD-G-111-026 (C.A.H.), DOH98-TD-G-111-015 (C.A.H.), DOH99-TD-G-111-028 (C.A.H.); National Health Research Institutes grants NHRI-PH-110-GP-01, NHRI-PH-110-GP-03; and the Ministry of Science and Technology grants MOST108-2314-B-400-038(C.A.H.), MOST109-2740-B-400-002(C.A.H.).

N.C. is supported by NIH grant 1R01HG010480. P.Y. is supported by Mayo Clinic Foundation Research Funds, NIH-CA77118 and CA80127. G.L. is supported is supported by the Alan Brown Chair and Lusi Wong Fund of the Princess Margaret Cancer Foundation. D.C.C. is supported by U01CA209414. O.Y.G. is supported by NIH R01 CA231141.

## Author contributions

Organized and designed the study: Q.L., J.S., N.R., J.C., S.J.C., N.C., K.S., K.M., T.K., M.T.L., C.I.A., O.Y.G., H.S., I-S.C., C.A.H., H.Shen. Conducted and supervised new genotyping for the project: K.S., K.M., T.K. Contributed to the design and execution of statistical analyses: J.S., J.C., S.J.C., N.R., Q.L., L.S., B.D.R., S.L, R.J.H., C.I.A., O.Y.G., N.C., IS.C., K.M.F., K.S., K.M., T.K. W.J.S., T.Z., C.B., M.J.M., I.C., H.Shen. Wrote the first draft: J.S., J.C., N.R., Q.L., K.S., K.M., T-Y.C., J.D., R.J.H., K.C., N.C., O.Y.G., C.A.H., S.J.C., C.I.A., H.S., T.K., H.Shen. Conducted epidemiology studies and contributed samples to GWAS and/or conducted initial genotyping: Q.L., M.T.L., B.A.B., W.H., N.E.C., BT.J., M.S., H.P., D.A., C.C., L.B., M.Y., A.H., B.H., J.Liu, B.Zhu, S.I.B., C.H.K., K.Wyatt, S.A.L., A.Chao, J.F.F.J., S.J.C., N.R., Z.Wang, C.L.W., J.C., C.W., W.T., D.Lin, SJ.A., XC.Z., J.S., YL.W., M.P.W., L.P.C., J.C.M.H., V.H.F.L., Z.H., K.M., J.Y.P., Jia.Liu, HS.J., J.E.C., Y.Y.C., H.N.K., MH.S., SS.K., YC.K., IJ.O., S.W.S., HI.Y., Y.T.K., YC.H., J.H.K., Y.H.K., J.S.S., Y.J.J., K.H.P., C.H.K., J.S.K., I.K.P., B.S., Jie.Liu, Z.W., S.C., J.Y., J.W., Y.Y., YT.G., D.L., J.YY.W., H.C., L.J., J.Z., G.J., K.F., Z.Y., B.Z., W.W., P.G., Q.H., X.L., Y.R., A.S., Y.L., Y.C., WY.L., W.Z., XO.S., Q.C., G.Y., B.Q., T.W., H.G., L.L., P.X., F.W., G.W., J.X., J.L., R.C.H.V., B.B., H.D.H., Junwen W., J.Wang, A.D.L.S., J.K.C.C., V.L.S., K.C., H.Z., H.D., C.A.H., T-Y.C., LH.C., I-S.C., CY.C., S.S.J., CH.C., GC.C., CF.H., YH.T., WC.W., KY.C., MS.H., WC.S., YM.C., CL.W., KC.C., CJ.Y., HH.H., FY.T., HC.L., CJ.C., PC.Y., K.S., T.K., H.K., S.M., H.H., K.G., Y.O., S.W., Y.Yatabe, M.T., R.H, A.T., Y.M., M.K., Y.K., Y.D., Y.Miyagi, H.N., T.Y., N.S., M.I., M.H., Y.N., K.T., K.W., K.Matsuda, Y.Murakami, K.S., K.T., Y.O., M.S., H.S., A.G., Y.M., T.H., M.K., K.O., H.S., J.D., H.M., M.Z., R.J.H., S.L., A.T., C.C., S.E.B., M.Johansson, A.R., HE.W., D.C.C., G.R., S.A., P.B., J.MK., J.K.F., S.S.S., L.L.M., H.Bö., G.L., A.A., L.A.K., S.ZN., K.G., M.J., A.C., JM.Y., P.L., M.B.S., M.C.A., C.I.A., A.G.S., R.H., M.R.S., O.Y.G., I.P.G., X.W., P.Y., J.Chang, M.Kobayashi, Y.Minamiya, K.Shimizu, M.Saito, Y.Ohtaki, K.Tanaka, J.Su, D.Lu, R.Houlston, S.Lam, A.Tardon, K.Grankvist, F.T., J.Wu, J.Li, H.Shen. All authors contributed to the writing and final review of the paper.

## Funding

## Competing interests
The authors declare no competing interests.

## Additional information

Jianxin Shi [1,146] ✉, Kouya Shiraishi [2,146], Jiyeon Choi [1,146], Keitaro Matsuo [3,146], Tzu-Yu Chen [4,146], Juncheng Dai [5,6,146], Rayjean J. Hung [7,146], Kexin Chen [8,146], Xiao-Ou Shu [9,146], Young Tae Kim [10], Maria Teresa Landi[1], Dongxin Lin[11], Wei Zheng [9], Zhihua Yin[12], Baosen Zhou[13], Bao Song[14], Jiucun Wang [15,16], Wei Jie Seow[1,17,18], Lei Song[1], I-Shou Chang [19], Wei Hu[1], Li-Hsin Chien[4], Qiuyin Cai [9], Yun-Chul Hong[20], Hee Nam Kim[21], Yi-Long Wu [22],

Maria Pik Wong[23], Brian Douglas Richardson[1,24], Karen M. Funderburk[1], Shilan Li[1,25], Tongwu Zhang[1], Charles Breeze[1], Zhaoming Wang[26], Batel Blechter[1], Bryan A. Bassig[1], Jin Hee Kim[27], Demetrius Albanes[1], Jason Y. Y. Wong[1], Min-Ho Shin[21], Lap Ping Chung[23], Yang Yang[28], She-Juan An[22], Hong Zheng[8], Yasushi Yatabe[29], Xu-Chao Zhang[22], Young-Chul Kim[30,31], Neil E. Caporaso[1], Jiang Chang[32], James Chung Man Ho[33], Michiaki Kubo[34], Yataro Daigo[35,36], Minsun Song[37], Yukihide Momozawa[34], Yoichiro Kamatani[38], Masashi Kobayashi[39], Kenichi Okubo[39], Takayuki Honda[40], Dean H. Hosgood[41], Hideo Kunitoh[42], Harsh Patel[1], Shun-ichi Watanabe[43], Yohei Miyagi[44], Haruhiko Nakayama[45], Shingo Matsumoto[46], Hidehito Horinouchi[43], Masahiro Tsuboi[47], Ryuji Hamamoto[48], Koichi Goto[46], Yuichiro Ohe[43], Atsushi Takahashi[38], Akiteru Goto[49], Yoshihiro Minamiya[50], Megumi Hara[51], Yuichiro Nishida[51], Kenji Takeuchi[52], Kenji Wakai[52], Koichi Matsuda[53], Yoshinori Murakami[54], Kimihiro Shimizu[55], Hiroyuki Suzuki[56], Motonobu Saito[57], Yoichi Ohtaki[58], Kazumi Tanaka[58], Tangchun Wu[59], Fusheng Wei[60], Hongji Dai[8], Mitchell J. Machiela[1], Jian Su[22], Yeul Hong Kim[61], In-Jae Oh[30,31], Victor Ho Fun Lee[62], Gee-Chen Chang[63,64,65,66], Ying-Huang Tsai[67,68], Kuan-Yu Chen[69], Ming-Shyan Huang[70], Wu-Chou Su[71], Yuh-Min Chen[72], Adeline Seow[17], Jae Yong Park[73], Sun-Seog Kweon[21,74], Kun-Chieh Chen[64], Yu-Tang Gao[75], Biyun Qian[8], Chen Wu[11], Daru Lu[15,16], Jianjun Liu[76,77], Ann G. Schwartz[78], Richard Houlston[79], Margaret R. Spitz[80], Ivan P. Gorlov[80], Xifeng Wu[81], Ping Yang[82], Stephen Lam[83], Adonina Tardon[84], Chu Chen[85], Stig E. Bojesen[86,87], Mattias Johansson[88], Angela Risch[89,90,91], Heike Bickeböller[92], Bu-Tian Ji[1], H-Erich Wichmann[93,94,95], David C. Christiani[96], Gadi Rennert[97], Susanne Arnold[98], Paul Brennan[88], James McKay[88], John K. Field[99], Sanjay S. Shete[100], Loic Le Marchand[101], Geoffrey Liu[102], Angeline Andrew[103], Lambertus A. Kiemeney[104], Shan Zienolddiny-Narui[105], Kjell Grankvist[106], Mikael Johansson[107], Angela Cox[108], Fiona Taylor[108], Jian-Min Yuan[109], Philip Lazarus[110], Matthew B. Schabath[111], Melinda C. Aldrich[112], Hyo-Sung Jeon[113], Shih Sheng Jiang[19], Jae Sook Sung[61], Chung-Hsing Chen[19], Chin-Fu Hsiao[4], Yoo Jin Jung[114], Huan Guo[115], Zhibin Hu[5], Laurie Burdett[1,116], Meredith Yeager[1,116], Amy Hutchinson[1,116], Belynda Hicks[1,116], Jia Liu[1,116], Bin Zhu[1,116], Sonja I. Berndt[1], Wei Wu[12], Junwen Wang[117,118], Yuqing Li[119], Jin Eun Choi[113], Kyong Hwa Park[61], Sook Whan Sung[120], Li Liu[121], Chang Hyun Kang[114], Wen-Chang Wang[122], Jun Xu[123], Peng Guan[12,124], Wen Tan[11], Chong-Jen Yu[125], Gong Yang[9], Alan Dart Loon Sihoe[126], Ying Chen[17], Yi Young Choi[113], Jun Suk Kim[127], Ho-Il Yoon[128], In Kyu Park[114], Ping Xu[129], Qincheng He[12], Chih-Liang Wang[130], Hsiao-Han Hung[19], Roel C. H. Vermeulen[131], Iona Cheng[132], Junjie Wu[15,16], Wei-Yen Lim[17], Fang-Yu Tsai[19], John K. C. Chan[133], Jihua Li[134], Hongyan Chen[15,16], Hsien-Chih Lin[4], Li Jin[15,16], Jie Liu[14], Norie Sawada[135], Taiki Yamaji[136], Kathleen Wyatt[1,116], Shengchao A. Li[1,116], Hongxia Ma[5,6], Meng Zhu[5,6], Zhehai Wang[14], Sensen Cheng[14], Xuelian Li[12,124], Yangwu Ren[12,124], Ann Chao[137], Motoki Iwasaki[135,136], Junjie Zhu[28], Gening Jiang[28], Ke Fei[28], Guoping Wu[60], Chih-Yi Chen[138,139], Chien-Jen Chen[140], Pan-Chyr Yang[141], Jinming Yu[14], Victoria L. Stevens[142], Joseph F. Fraumeni Jr[1], Nilanjan Chatterjee[1,143,144,147], Olga Y. Gorlova[80,145,147], Chao Agnes Hsiung[4,147], Christopher I. Amos[80,145,147], Hongbing Shen[5,6,147], Stephen J. Chanock[1,147], Nathaniel Rothman[1,147], Takashi Kohno[2,147] & Qing Lan[1,147] ✉

[1]Division of Cancer Epidemiology and Genetics, National Cancer Institute, Rockville, MD, USA. [2]Division of Genome Biology, National Cancer Research Institute, Tokyo, Japan. [3]Division of Cancer Epidemiology and Prevention, Aichi Cancer Center Research Institute, Nagoya, Japan. [4]Institute of Population Health Sciences, National Health Research Institutes, Zhunan, Taiwan. [5]Department of Epidemiology, School of Public Health, Nanjing Medical University, Nanjing, China. [6]Jiangsu Key Lab of Cancer Biomarkers, Prevention and Treatment, Collaborative Innovation Center for Cancer Medicine, Nanjing Medical University, Nanjing, China. [7]Prosserman Centre for Population Health Research, Lunenfeld-Tanenbaum Research Institute, Sinai Health, Toronto, ON, Canada. [8]Department of Epidemiology and Biostatistics, National Clinical Research Center for Cancer, Key Laboratory of Molecular Cancer Epidemiology of Tianjin, Tianjin Medical University Cancer Institute and Hospital, Tianjin Medical University, Tianjin, China. [9]Division of Epidemiology, Department of Medicine, Vanderbilt University Medical Center and Vanderbilt-Ingram Cancer Center, Nashville, TN, USA. [10]Cancer Research Institute, Seoul National University College of Medicine, Seoul, Republic of Korea. [11]Department of Etiology & Carcinogenesis and State Key Laboratory of Molecular Oncology, Cancer Institute and Hospital, Chinese Academy of Medical Sciences and Peking Union Medical College, Beijing, China. [12]Department of Epidemiology, School of Public Health, China Medical University, Shenyang, China. [13]Department of Clinical Epidemiology and Center of Evidence Based Medicine, The First Hospital of China Medical University, Shenyang, China. [14]Department of Oncology, Shandong Cancer Hospital and Institute, Shandong Academy of Medical Sciences, Jinan, China. [15]Ministry of Education Key Laboratory of Contemporary Anthropology, School of Life Sciences,  Fudan University, Shanghai, China. [16]State Key Laboratory of Genetic Engineering, School of Life Sciences, Fudan University, Shanghai, China. [17]Saw Swee Hock School of Public Health, National University of Singapore, Singapore, Singapore. [18]Department of Medicine, Yong Loo Lin School of Medicine, National University of Singapore and National University Health System, Singapore, Singapore. [19]National Institute of Cancer Research, National Health Research Institutes, Zhunan, Taiwan. [20]Department of Preventive Medicine, Seoul National University College of Medicine, Seoul, Republic of Korea. [21]Department of Preventive Medicine, Chonnam National University Medical School, Gwangju, Republic of Korea. [22]Guangdong Lung Cancer Institute, Medical Research Center and Cancer Center of Guangdong Provincial People's Hospital, Guangdong Academy of Medical Sciences, Guangzhou, China. [23]Department of Pathology, Queen Mary Hospital, Hong Kong, Hong Kong. [24]Department of Biostatistics, Gillings School of Global Public Health, University of North Carolina, Chapel Hill, NC, USA. [25]Department of Biostatistics, Bioinformatics & Biomathematics, Georgetown University Medical Center, Washington, DC, USA. [26]Department of Computational Biology, St. Jude Children's Research Hospital, Memphis, TN, USA. [27]Department of Environmental Health, Graduate School of Public Health, Seoul National University, Seoul, Republic of Korea. [28]Shanghai Pulmonary Hospital, Shanghai, China. [29]Department of Pathology and Clinical Laboratories, National Cancer Center Hospital, Tokyo, Japan. [30]Lung and Esophageal Cancer Clinic, Chonnam National University Hwasun Hospital, Hwasuneup, Republic of Korea. [31]Department of Internal Medicine, Chonnam National Univerisity Medical School, Gwangju, Republic of Korea. [32]Department of Etiology & Carcinogenesis, Cancer Institute

and Hospital, Chinese Academy of Medical Sciences and Peking Union Medical College, Beijing, China. [33]Department of Medicine, The University of Hong Kong, Queen Mary Hospital, Hong Kong, Hong Kong. [34]Laboratory for Genotyping Development, RIKEN Center for Integrative Medical Sciences, Yokohama, Japan. [35]Center for Antibody and Vaccine Therapy, Research Hospital, Institute of Medical Science, The University of Tokyo, Tokyo, Japan. [36]Department of Medical Oncology and Cancer Center, and Center for Advanced Medicine against Cancer, Shiga University of Medical Science, Shiga, Japan. [37]Department of Statistics & Research Institute of Natural Sciences, Sookmyung Women's University, Seoul, Republic of Korea. [38]Laboratory for Statistical Analysis, RIKEN Center for Integrative Medical Sciences, Yokohama, Japan. [39]Department of Thoracic Surgery, Tokyo Medical and Dental University, Tokyo, Japan. [40]Department of Respiratory Medicine, Tokyo Medical and Dental University, Tokyo, Japan. [41]Department of Epidemiology and Population Health, Albert Einstein College of Medicine, New York, NY, USA. [42]Department of Medical Oncology, Japanese Red Cross Medical Center, Tokyo, Japan. [43]Department of Thoracic Surgery, National Cancer Center Hospital, Tokyo, Japan. [44]Molecular Pathology and Genetics Division, Kanagawa Cancer Center Research Institute, Yokohama, Japan. [45]Department of Thoracic Surgery, Kanagawa Cancer Center, Yokohama, Japan. [46]Department of Thoracic Oncology, National Cancer Center Hospital East, Kashiwa, Japan. [47]Department of Thoracic Surgery, National Cancer Center Hospital East, Kashiwa, Japan. [48]Division of Medical AI Research and Development, National Cancer Center Research Institute, Tokyo, Japan. [49]Department of Cellular and Organ Pathology, Graduate School of Medicine, Akita University, Akita, Japan. [50]Department of Thoracic Surgery, Graduate School of Medicine, Akita University, Akita, Japan. [51]Department of Preventive Medicine, Faculty of Medicine, Saga University, Saga, Japan. [52]Department of Preventive Medicine, Nagoya University Graduate School of Medicine, Nagoya, Japan. [53]Laboratory of Clinical Genome Sequencing, Department of Computational Biology and Medical Science, Graduate School of Frontier Sciences, The University of Tokyo, Tokyo, Japan. [54]Division of Molecular Pathology, Institute of Medical Science, The University of Tokyo, Tokyo, Japan. [55]Department of Surgery, Division of General Thoracic Surgery, Shinshu University School of Medicine Asahi, Nagano, Japan. [56]Department of Chest Surgery, Fukushima Medical University School of Medicine, Fukushima, Japan. [57]Department of Gastrointestinal Tract Surgery, Fukushima Medical University School of Medicine, Fukushima, Japan. [58]Department of Integrative center of General Surgery, Gunma University Hospital, Gunma, Japan. [59]Institute of Occupational Medicine and Ministry of Education Key Lab for Environment and Health, School of Public Health, Huazhong University of Science and Technology, Wuhan, China. [60]China National Environmental Monitoring Center, Beijing, China. [61]Department of Internal Medicine, Division of Oncology/Hematology, College of Medicine, Korea University Anam Hospital, Seoul, Republic of Korea. [62]Department of Clinical Oncology, The University of Hong Kong, Queen Mary Hospital, Hong Kong, Hong Kong. [63]School of Medicine and Institute of Medicine, Chung Shan Medical University, Taichung, Taiwan. [64]Department of Internal Medicine, Division of Pulmonary Medicine, Chung Shan Medical University Hospital, Taichung, Taiwan. [65]Institute of Biomedical Sciences, National Chung Hsing University, Taichung, Taiwan. [66]Department of Internal Medicine, Division of Chest Medicine, Taichung Veterans General Hospital, Taichung, Taiwan. [67]Department of Respiratory Therapy, Chang Gung University, Taoyuan, Taiwan. [68]Department of Pulmonary and Critical Care, Xiamen Chang Gung Hospital, Xiamen, China. [69]Department of Internal Medicine, National Taiwan University Hospital and College of Medicine, Taipei, Taiwan. [70]Department of Internal Medicine, E-Da Cancer Hospital, I-Shou University and Kaohsiung Medical University, Kaohsiung, Taiwan. [71]Department of Oncology, National Cheng Kung University Hospital, College of Medicine, National Cheng Kung University, Tainan, Taiwan. [72]Department of Chest Medicine, Taipei Veterans General Hospital, and school of Medicine, National Yang Ming Chiao Tung University, Taipei, Taiwan. [73]Lung Cancer Center, Kyungpook National University Medical Center, Daegu, Republic of Korea. [74]Jeonnam Regional Cancer Center, Chonnam National University, Hwasun, Republic of Korea. [75]Department of Epidemiology, Shanghai Cancer Institute, Shanghai, China. [76]Genome Institute of Singapore, Agency of Science, Technology and Research, Singapore, Singapore. [77]Yong Loo Lin School of Medicine, National University of Singapore, Singapore, Singapore. [78]Karmanos Cancer Institute, Detroit, MI, USA. [79]Division of Genetics and Epidemiology, Institute of Cancer Research, London, UK. [80]Department of Medicine, Section of Epidemiology and Population Science, Institute for Clinical and Translational Research, Houston, TX, USA. [81]School of Medicine, Zhejiang University, Hangzhou, Zhejiang, China. [82]Department of Health Sciences Research, Mayo Clinic, Scottsdale, AZ, USA. [83]British Columbia Cancer Agency, Vancouver, BC, Canada. [84]IUOPA, University of Oviedo and CIBERESP, Madrid, Spain. [85]Public Health Sciences Division, Fred Hutchinson Cancer Center, Seattle, WA, USA. [86]Faculty of Health and Medical Sciences, University of Copenhagen, Copenhagen, Denmark. [87]Department of Clinical Biochemistry, Herlev and Gentofte Hospital, Copenhagen University Hospital, Copenhagen, Denmark. [88]International Agency for Research on Cancer (IARC/WHO), Lyon, France. [89]German Cancer Research Center (DKFZ), Heidelberg, Germany. [90]Translational Lung Research Center Heidelberg (TLRC-H), Member of the German Center for Lung Research (DZL), Heidelberg, Germany. [91]University of Salzburg and Cancer Cluster Salzburg, Salzburg, Austria. [92]University Medical Center Goettingen, Goettingen, Germany. [93]Institute of Medical Informatics, Biometry and Epidemiology, Ludwig Maximilians University, Munich, Germany. [94]Helmholtz Center Munich, Institute of Epidemiology, Munich, Germany. [95]Institute of Medical Statistics and Epidemiology, Technical University Munich, Munich, Germany. [96]Harvard TH Chan School of Public Health, Boston, MA, USA. [97]Carmel Medical Center, Haifa, Israel. [98]Markey Cancer Center, Lexington, KY, USA. [99]Liverpool University, Liverpool, UK. [100]The University of Texas MD Anderson Cancer Center, Houston, TX, USA. [101]Epidemiology Program, University of Hawaii Cancer Center, Honolulu, HI, USA. [102]Princess Margaret Cancer Center, Toronto, ON, Canada. [103]Norris Cotton Cancer Center, Lebanon, NH, USA. [104]Radboud University Medical Center, Nijmegen, Netherlands. [105]National Institute of Occupational Health, Oslo, Norway. [106]Department of Medical Biosciences, Umeå University, Umeå, Sweden. [107]Department of Radiation Sciences, Umeå University, Umeå, Sweden. [108]University of Sheffield, Sheffield, UK. [109]UPMC Hillman Cancer Center and Department of Epidemiology, School of Public Health, University of Pittsburgh, Pittsburgh, PA, USA. [110]Washington State University College of Pharmacy, Spokane, WA, USA. [111]Department of Cancer Epidemiology, H. Lee Moffitt Cancer Center and Research Institute, Tampa, FL, USA. [112]Department of Thoracic Surgery, Division of Epidemiology, Vanderbilt University Medical Center, Nashville, TN, USA. [113]Cancer Research Center, Kyungpook National University Medical Center, Daegu, Republic of Korea. [114]Department of Thoracic and Cardiovascular Surgery, Cancer Research Institute, Seoul National University College of Medicine, Seoul, Republic of Korea. [115]Department of Occupational and Environmental Health and Ministry of Education Key Lab for Environment and Health, School of Public Health, Tongji Medical College, Huazhong University of Science and Technology, Wuhan, China. [116]Cancer Genomics Research Laboratory, Leidos Biomedical Research Inc., Rockville, MD, USA. [117]Department of Biochemistry, Li Ka Shing (LKS) Faculty of Medicine, The University of Hong Kong, Hong Kong, China. [118]Centre for Genomic Sciences, Li Ka Shing (LKS) Faculty of Medicine, The University of Hong Kong, Hong Kong, China. [119]Department of Human Genetics, Genome Institute of Singapore, Singapore, Singapore. [120]Department of Thoracic and Cardiovascular Surgery, Seoul National University Bundang Hospital, Seongnam, Republic of Korea. [121]Department of Oncology, Cancer Center, Union Hospital, Huazhong University of Science and Technology, Wuhan, China. [122]The Ph.D. Program for Translational Medicine, College of Medical Science and Technology, Taipei Medical University, Taipei, Taiwan. [123]School of Public Health, Li Ka Shing (LKS) Faculty of Medicine, The University of Hong Kong, Hong Kong, China. [124]Key Laboratory of Cancer Etiology and Intervention, University of Liaoning Province, Shenyang, China. [125]Department of Internal Medicine, National Taiwan University Hospital Hsin-Chu Branch, Hsinchu, Taiwan. [126]Gleneagles Hong Kong Hospital, Hong Kong, China. [127]Department of Internal Medicine, Division of Medical Oncology, College of Medicine, Korea University Guro Hospital, Seoul, Republic of Korea. [128]Department of Internal Medicine, Seoul National University Bundang Hospital, Seongnam, Republic of Korea. [129]Department of Oncology, Wuhan Iron and Steel (Group) Corporation Staff-Worker Hospital, Wuhan, China. [130]Department of Pulmonary and Critical Care, Chang Gung Memorial Hospital, Taoyuan, Taiwan. [131]Division of Environmental

Epidemiology, Institute for Risk Assessment Sciences (IRAS), Utrecht University, Utrecht, The Netherlands. [132]Department of Epidemiology and Biostatistics, University of California, San Francisco, San Francisco, CA, USA. [133]Department of Pathology, Queen Elizabeth Hospital, Hong Kong, China. [134]Qujing Center for Diseases Control and Prevention, Qujing, China. [135]Division of Cohort Research, National Cancer Center Institute for Cancer Control, National Cancer Center, Tokyo, Japan. [136]Division of Epidemiology, National Cancer Center Institute for Cancer Control, National Cancer Center, Tokyo, Japan. [137]Center for Global Health, National Cancer Institute, Bethesda, MD, USA. [138]Institute of Medicine, Chung Shan Medical University, Taichung, Taiwan. [139]Division of Thoracic Surgery, Department of Surgery, Chung Shan Medical University Hospital, Taichung, Taiwan. [140]Genomic Research Center, Academia Sinica, Taipei, Taiwan. [141]Department of Internal Medicine, National Taiwan University Hospital, Taipei, Taiwan. [142]Laboratory Services, American Cancer Society, Atlanta, GA, USA. [143]Department of Oncology, School of Medicine, Johns Hopkins University, Baltimore, MD, USA. [144]Department of Biostatistics, Johns Hopkins Bloomberg School of Public Health, Baltimore, MD, USA. [145]Dan L Duncan Comprehensive Cancer Center, Baylor College of Medicine, Houston, TX, USA. [146]These authors contributed equally: Jianxin Shi, Kouya Shiraishi, Jiyeon Choi, Keitaro Matsuo, Tzu-Yu Chen, Juncheng Dai, Rayjean J Hung, Kexin Chen, Xiao-Ou Shu [147]These authors jointly supervised this work: Nilanjan Chatterjee, Olga Y. Gorlova, Chao Agnes Hsiung, Christopher I. Amos, Hongbing Shen, Stephen J. Chanock, Nathaniel Rothman, Takashi Kohno, Qing Lan. ✉e-mail: jianxin.shi@nih.gov; qingl@mail.nih.gov

