## [Peer Review File · Nature Communications]

Genome-wide association study of lung adenocarcinoma in East Asia and comparison with a European populationREVIEWER COMMENTS

Reviewer #1 (Remarks to the Author):

Supp Fig 9 shows <20% heritability of LUAD, and esp EA never-smokers <10%. Can you discuss the implications of this? I'm surprised that EA never-smokers have so little heritability, could this be because cooking fuel exposures are predominant?

Do you have cooking fuel exposure information, which may be the major source of LUAD in EA women? If not, how do you think this lack of information affects your findings?

For MR analysis, the MR Egger method is stat insignificant - what to make of this? What are the weaknesses of this analysis? Can you have a discussion sentence about why longer not shorter telomeres are plausibly associated with LUAD?

The 95%CI for GENESIS overlaps 0: (~0,~4400) variants assoc with LUAD. There is so much variation, is this a reliable analysis? How does this affect your figure 5 on AUC - what is the pointwise CI for each estimate, does it overlap AUC=0.5?

There are no supplemental tables 6-11, nor 15 comparing EA and EUR population PRS, nor 16.

For PRS*smoking interaction, why is the variance of δ_i equal to δ_i ?

What is the mean and variance of the LUAD PRS in EA and EUR populations? Because the SNP effects are different, I expect there to be different means and variances and PRS will be collinear with any race/ethnicity variables. Can you comment on the potential clinical use of a PRS that contains race/ethnicity information, given concerns about the fairness of use of race/ethnicity in models?

Reviewer #2 (Remarks to the Author):

This manuscript presents findings from a GWAS meta-analysis lung adenocarcinoma of East Asian (EA) ancestry (21,658 cases and 150,676 controls) and a multi-ancestry meta-analysis with a previously

published European ancestry GWAS. I commend the authors for the scope and amount of work that went into this paper, but I'm afraid that in conducting this multitude of analyses, the paper loses focus. At times it is difficult to keep track of the data sources, criteria for statistical significance, novelty, colocalization etc. The different components of the paper are not well synthesized, giving the impression that because so many analyses were done, many of them were conducted rather superficially. This is especially true for the Mendelian Randomization analysis and TWAS, which seem like add ons. The TWAS has some interesting findings, but they are not fully explored. The PRS gene-environment analysis glosses over the difficulty with disentangling histology-specific susceptibility signals from differences in SNP effects on smoking.

It feels like this was a missed opportunity to really dig into ancestry differences. Most comparisons are based on statistical significance, which is a problem when there is a sample size imbalance across ancestries. There is little discussion of allelic/effect size heterogeneity and no attempt to develop an ancestry informed PRS or compare different PRS approaches. There is also no consideration of ancestry in the eQTL/TWAS component or MR analysis. Overall, despite so many types of analyses conducted and datasets leveraged, the contribution of this paper to the literature seems incremental.

Specific comments:

- I suggest reconsidering the use of the term "trans" in this manuscript, please see PMID: 34741159 for discussion of the misuse of this term in genetic association studies. Multi-ancestry or cross-ancestry would be more appropriate.

- Please clarify if analyses were restricted to variants with $MAF \geq 0.01$ in both EAS and EUR populations, or if SNPs that were common in only one population were included.

- The cross-ancestry comparisons seem limited and primarily based on genome-wide significance. Visual inspection of regional LD plots was also performed, but it's not clear if this was systematically conducted for all variants. For example, what does "significantly different" mean here: "the majority of SNP associations initially identified in EA populations showed no evidence of association within EUR populations and most were significantly different."

- The section titled "Comparing the genetics of LUAD in EA and EUR populations" has vague statements, such as "majority of SNPs" or "several SNPs". Please provide the specific numbers of variants and the denominator for each comparison.

- The authors conclude that “The lack of association in EUR populations did not seem to be driven by low MAF or lower statistical power” – so then what could be driving these differences? Is it possible that there are different lead/index variants at some of these regions that were somehow not captured in the analysis?

- POPCORN analysis of cross-population heterogeneity seems inconclusive and limited

statistical power is given as the reason on page 14. This seems to contradict claims elsewhere in the paper. Or perhaps POPCORN is simply not the right method to use for looking at heterogeneity?

- The criteria for novelty are not explicitly articulated. It would be helpful to distinguish novel loci from novel variants and provide some nuance around what is considered “novel” for a specific population or novel for LUAD histology. For instance, on page 9, none of the 5p15.33 variants appear to be truly novel: rs13167280 has been previously reported in several studies in European (PMID: 30059977; 28604730 22523397) and East Asian populations (PMID: 31326317) and rs62332591 has also been previously reported in Europeans (PMID: 28604730) and African Americans (PMID: 23221128). Furthermore, I wouldn’t characterize $R^2 < 0.25$ as low LD, particularly for detecting “novel” associations in such a well-characterized lung cancer susceptibility region as 5p15.33.

- Why was such an old version of GTEx used for the tissue-specific analysis? It’s possible that lack of enrichment signal could be due to the fact that GTEx v6 was used

- The choice of 50 SNPs around the lead eQTL for eCAVIAR is not well motivated or consistent with other fine-mapping analyses in this manuscript. Please consider using a more standard, window-based approach, such as +/- 500kb. Similarly, the reasoning behind the extremely low threshold for the colocalization posterior probability (>0.01) is unclear. Please describe how this threshold was chosen and why this is appropriate for this analysis. Also, please distinguish between regional posterior probabilities and SNP-specific posterior probabilities.

- Comparisons of the top vs. bottom decile should be avoided, for PRS and especially for MR. This presents a distorted view of the data that is only designed to elicit a large effect size by comparing two extremes. This is not an informative comparison because typically the goal of the PRS is to improve our ability to identify high-risk individuals in the general population, so comparisons should be with respect to the average risk group (40-60th percentile or 20-80th percentile). This is even more unnecessary for MR and hinders comparisons with other MR studies of lung cancer and TL. There is no need to go through the complicated conversion, just report the OR that corresponds to the increase in lung cancer risk per unit increase in telomere length and report what that “unit” corresponds to (which depends on how TL was operationalized in the TOPMed GWAS)

- MR analyses have a number of issues. Firstly, the instrument selection procedure is not described at all. How were these SNPs selected? What was the p-value and LD threshold? Given the differences in ancestry between the TOPMed telomere length GWAS and the present analysis, did the authors make any efforts to ensure that these SNPs and corresponding effect sizes were appropriate? Was there evidence of ancestry-related heterogeneity for any of these variants?
- Basic information about the MR analysis is missing and there are no MR diagnostics presented. Was the inverse variance weighted (IVW) analysis done using a fixed effects or multiplicative random effects model? Please clarify
- Why were MR Egger, weighted median, and weighted mode chosen as additional MR methods? Was there any evidence of directional pleiotropy to justify the use of MR Egger? If the MR Egger intercept is not significantly different from zero, then there is no need to use MR Egger for estimating causal effects
- Was there any evidence of heterogeneity in the SNP-specific causal effect estimates? Presence of heterogeneity would indicate potential for balanced horizontal pleiotropy, which can be addressed using appropriate methods (random effects IVW or MR PRESSO).
- Weighted median and weighted mode are the same general class of MR estimator, these are consensus-based methods that assume that the true causal effect is the one estimated by the largest number of instruments. Including both is OK but a bit redundant. I would instead suggest including another MR estimation method that takes a different approach to pleiotropy detection and correction
- TWAS findings are not explored. For instance, is there eQTL sharing with other genes in the region, are ELF5 and FADS1 likely to be the main causal genes?
- The development of transcriptome imputation models using LCTCNS eQTL data is a strength of this paper, but it gets lost in everything else. Are the models publicly available? How do they compare with GTEx data in terms of the prediction performance and number of genes captured? What is the number of overlapping genes? Does TWAS using LCTCNS models identify genes that would not be detected using ancestry-discordant models?
- The Discussion seems very brief and only focuses on selected findings, doesn't provide sufficient context for the other analyses and doesn't discuss any limitations or challenges with cross-ancestry comparisons (beyond sample size)

Reviewer #3 (Remarks to the Author):

Shi and colleagues conducted a two-stage genome-wide association study of lung adenocarcinoma in individuals of East Asian (EA) ancestry, including 11,753 cases and 30,562 controls from four studies in the discovery phase and 9,905 cases and 120,114 controls from Japan in the replication phase, through which they identified 14 novel genetic risk variants. They performed post-GWAS functional characterization of the identified risk loci, encompassing eQTL colocalization and transcriptome-wide association analyses. In addition, they combined data from EA (11,753 cases and 30,562 controls) and European (EUR; 11,273 cases and 55,483 controls) populations in a trans-ethnic GWAS meta-analysis, identifying four genetic variants that confer similar risk effects for lung adenocarcinoma in both populations. They further compared risk effect sizes of both novel and previously known genetic risk variants for lung adenocarcinoma between EA and EUR populations. Additional analyses revealed that all known genetic variants combined as a polygenic risk score (PRS) and some individual genetic risk variants were more strongly associated with lung adenocarcinoma risk in individuals who never smoked than those who smoked; that about 2,275 genetic variants may be independently associated with lung adenocarcinoma in EA populations; and that expanding GWAS to include 70,000 cases and 70,000+ controls could further improve knowledge about the genetic architecture of lung adenocarcinoma, including genetic interactions with tobacco smoking and other environmental risk factors. Findings from their extensive work reinforce the importance of conducting genetic association studies of lung adenocarcinoma in non-EUR ancestral populations. Overall, the study rationale, methods, and results are clearly described and presented, and only minor comments are suggested to improve this well-written manuscript.

1. The discovery phase included individuals of diverse EA populations from mainland China, Hong Kong, Singapore, Taiwan, South Korea, and Japan, whereas the replication phase included individuals from Japan only. To what extent could differences in the composition of the discovery and replication sets have influenced the identification and confirmation of novel loci for lung adenocarcinoma, given that genetic differences do exist between EA ethnic groups? If this design choice was not ideal, it should be acknowledged as a study limitation.

2. Please clarify why no replication data were available for the variant identified on chromosome 7 (rs4268071). Of all the novel variants identified, this one is the least common (EAF=4%). Could this be a chance finding?

3. For the PRS-smoking interaction analysis, the population examined does not appear to be specified. Please clarify whether this analysis was conducted on the EA population only or both the EA and EUR populations separately. Also, it would be informative if an explanation was offered as to why the PRS showed a significantly stronger association among those with no smoking history.

4. The text refers to Supplementary Table 8 only, rather than Supplementary Tables 8A, 8B, 8C, and 8D.

5. Suggest reordering the presentation of the methods to follow that of the results.

6. Please use non-stigmatizing language (e.g., individuals with a smoking history, instead of smokers) whenever possible.

Reviewer #1 (Remarks to the Author):

Supp Fig 9 shows <20% heritability of LUAD, and esp EA never-smokers <10%. Can you discuss the implications of this? I'm surprised that EA never-smokers have so little heritability, could this be because cooking fuel exposures are predominant?

Response: Compared to other cancers, heritability of lung cancer, including LUAD, is relatively modest (18% for lung cancer compared to 33% for overall cancer based on the NorTwinCan Cohort study, Mucci *et al.*, *JAMA*, 2016; OR=1.7 (95% CI=1.49-1.94) for a positive family history based on a recent systematic meta-analysis, Ang *et al.*, *Lung Cancer*, 2020; for never smoking females in EA, estimated OR in recent studies ranged from 1.08 (0.66-1.77) to 2.78 (1.57-4.90), Yoshida *et al.*, *Tohoku J. Exp. Med.*, 2019, Yin *et al.*, *Scientific Reports*, 2021). Moreover, the reported heritability estimate in our manuscript is an array-based heritability that is attributed to the common variants with minor allele frequency 1% or higher. Additional heritability that could be potentially explained by rare variants may not be reflected in our assessment.

As the reviewer suggested, multiple environmental factors including cooking fuel exposures may play a role in lung cancer etiology in never-smokers, although this varies by geographic regions and living in urban vs. rural environments. However, even in countries where, for example, coal was the primary fuel exposure, cooking fuel exposure was not a major risk factor for LUAD in EA women. In our recent paper (Blechter *et al.*, *Environment International*, 2021), we estimated that ever-coal use had only a modest OR=1.29 (95% CI=1.01-1.64) for LUAD risk in East Asian never-smokers.

Do you have cooking fuel exposure information, which may be the major source of LUAD in EA women? If not, how do you think this lack of information affects your findings?

Response: Cooking fuel exposure is not a major source of LUAD in EA women. Please see the response above. A cooking fuel variable was available for a subgroup of 1,067 controls in our discovery set and we were able to use these data to evaluate the extent to which not having this information for the entire data set would have adversely affected our findings. Among these 1,067 subjects, risk variants were not associated with coal-use individually ($P > 0.05$ for all variants) or collectively ($P = 0.58$, polygenic risk score defined based on 25 independent susceptibility variants in Supplementary Table 4), which is as expected. Thus, coal use could not confound the association between LUAD and these risk variants.

Further, cooking fuel practices in mainland China have had a different pattern due in part to greater availability of coal and also less rapid industrialization compared to other centers in our study, i.e., Japan, South Korea, Singapore, Hong Kong, and Taiwan. We analyzed the genetic association of LUAD stratified by country of origin as a proxy for cooking practice differences, i.e., within Mainland China vs. outside Mainland China, and found that ORs for all 28 risk variants were similar (updated Supplementary Table 5, ORs correlated at $R = 0.955$; $P > 0.1$ for testing the heterogeneity of ORs for all variants between Mainland China and non-Mainland China, Line 290-292 in main text).

In summary, we found that LUAD risk variants were not associated with coal use, or cooking fuel, in our data and thus lack of cooking fuel information would not have confounded our results. In addition, the high concordance of ORs between Mainland China mainland and non-mainland China suggested that differential exposures may not have had a substantial effect on genetic effects.

For MR analysis, the MR Egger method is stat insignificant - what to make of this? What are the weaknesses of this analysis? Can you have a discussion sentence about why longer not shorter telomeres are plausibly associated with LUAD?

Response: We appreciate the comments from the reviewer and related comments from the second reviewer. We previously used this approach as a sensitivity analysis in the presence of horizontal pleiotropy. Reduced significance in MR Egger may suggest the existence of horizontal pleiotropy. Following the second reviewer's suggestion, we now use MR-PRESSO (Verbanck *et al.*, *Nat. Genet.*, 2018) as the primary tool for MR analysis. This approach uses a formal test framework to (1) detect the presence of horizontal pleiotropy, (2) detect variant outliers, (3) evaluate distortion, and (4) re-estimate causal effect sizes after removing potentially problematic variants. According to simulations, this approach is best suited when horizontal pleiotropy occurs in < 50% of instruments. In our real data analyses (six analyses, smokers/ never-smokers/ both in EUR/EA), this approach did identify 5-7 outlier variants. The causal effects were statistically significant after removing these outliers (see updated Supplementary Table 15).

It is biologically plausible that longer telomere length is associated with risk of LUAD. Cell-based and animal studies indicated that longer telomeres confer longer replicative potential (i.e., increased number of replication and increased lifespan of cells) which could allow proliferation of cells with genomic instability and abnormalities and lead to tumor formation and progression (McNally, Luncsford, and Armanios, *JCI*, 2019).

The 95%CI for GENESIS overlaps 0: (~0,~4400) variants assoc with LUAD. There is so much variation, is this a reliable analysis? How does this affect your figure 5 on AUC - what is the pointwise CI for each estimate, does it overlap AUC=0.5?

Response: We agree with the reviewer that, given the current sample size, estimating the number of potentially truly associated variants is not very precise, which will be improved with expanded GWAS in this population in the future. However, we can more precisely estimate other overall important metrics, including heritability ($h^2=9.9\%$ with standard error or s.e. = 2.2% using LDSC) and AUC of the PRS using a training GWAS data of a specified sample size. While the original GENESIS package (Zhang *et al.*, *Nature Genetics*, 2019) did not provide the option to estimate s.e. of the AUC, we developed a resampling approach for this purpose. In our data, estimated s.e. ranged from 1.6% to 2.4% and thus AUCs do not overlap with 50%. Technical details are included in Methods (Line 722-741). For some AUC values in the main text, 95% CI was added (Line 442 - 445):

“The AUC is predicted to be 60.7% (95% CI = 56.6%, 64.8%) at the current sample size and will increase to 66.9% (95% CI = 62.5%, 71.3%) when the sample size increases to 70,000 cases with one control per case and 68.4% (95% CI = 64.0%, 72.8%) with 1,000,000 controls. Of note, even a small increase of AUC value for a PRS can help identify many more subjects at risk⁷⁰. ”

There are no supplemental tables 6-11, nor 15 comparing EA and EUR population PRS, nor 16.

Response: Supplementary Figures and some Supplementary Tables (1-4, 6, 8, 12, 13, 15) were combined as a PDF file in the submitted manuscript. Other Supplementary Tables are submitted as one Excel file.

*For PRS*smoking interaction, why is the variance of delta_i equal to delta_i?*

Response: In our previous analysis, we assumed that genotypic variances were similar across cases and controls and across smokers and never-smokers, which was verified empirically. To make the analysis more robust and straightforward, we have followed the comment to modify the algorithm to use group-specific genotypic variances (see also below). Results are very similar. We updated Methods (Line 713 - 716) and Figure 4 accordingly.

“Assuming SNPs are independent, we derive $Z = \sum_{t=1}^T (w_t^1 z_t^1 - w_t^0 z_t^0)$, where z_t^s is the z-score for testing association for SNP t in subjects with smoking status s . The weight is given as

$$w_t^s = \frac{\beta_t \sqrt{\frac{(\sigma_t^{s+})^2}{n_+^s} + \frac{(\sigma_t^{s-})^2}{n_-^s}}}{\sqrt{\sum_{t=1}^T \beta_t^2 \left(\frac{(\sigma_t^{1+})^2}{n_+^1} + \frac{(\sigma_t^{1-})^2}{n_-^1} + \frac{(\sigma_t^{0+})^2}{n_+^0} + \frac{(\sigma_t^{0-})^2}{n_-^0} \right)}}$$

Here, $(\sigma_t^{s+})^2$ and $(\sigma_t^{s-})^2$ are the genotypic variance for SNP t in cases and controls, respectively.”

What is the mean and variance of the LUAD PRS in EA and EUR populations? Because the SNP effects are different, I expect there to be different means and variances and PRS will be collinear with any race/ethnicity variables. Can you comment on the potential clinical use of a PRS that contains race/ethnicity information, given concerns about the fairness of use of race/ethnicity in models?

Response: For EA populations, PRS was calculated based on 25 independent variants that achieved genome-wide significance in EA data set (Supplementary Table 4). For EUR populations, PRS was calculated based on 14 variants that achieved genome-wide significance in our EUR data set. Minor alleles were coded as 1. Mean = 3.12 and s.d. = 0.46 for EA; mean=1.51 and s.d.=0.32 for EUR. This is not unexpected, because PRS is not comparable between the two populations as they are based on different sets of variants. We agree that the future translational use of a PRS in conjunction with other risk factors within a comprehensive

and validated risk stratification model would need to be done in a careful and sensitive manner.

Reviewer #2 (Remarks to the Author):

This manuscript presents findings from a GWAS meta-analysis lung adenocarcinoma of East Asian (EA) ancestry (21,658 cases and 150,676 controls) and a multi-ancestry meta-analysis with a previously published European ancestry GWAS. I commend the authors for the scope and amount of work that went into this paper, but I'm afraid that in conducting this multitude of analyses, the paper loses focus. At times it is difficult to keep track of the data sources, criteria for statistical significance, novelty, colocalization etc. The different components of the paper are not well synthesized, giving the impression that because so many analyses were done, many of them were conducted rather superficially. This is especially true for the Mendelian Randomization analysis and TWAS, which seem like add ons. The TWAS has some interesting findings, but they are not fully explored. The PRS gene-environment analysis glosses over the difficulty with disentangling histology-specific susceptibility signals from differences in SNP effects on smoking.

It feels like this was a missed opportunity to really dig into ancestry differences. Most comparisons are based on statistical significance, which is a problem when there is a sample size imbalance across ancestries. There is little discussion of allelic/effect size heterogeneity and no attempt to develop an ancestry informed PRS or compare different PRS approaches. There is also no consideration of ancestry in the eQTL/TWAS component or MR analysis. Overall, despite so many types of analyses conducted and datasets leveraged, the contribution of this paper to the literature seems incremental.

Response: We have carefully addressed all specific comments the reviewer has provided and would also like to respond to the overall comments that have been made here that we were not asked to respond to subsequently.

Through our substantial new analyses and revisions of the manuscript in the Methods, Results, and Discussion, we believe that the manuscript is substantially improved and better focused, as well as enhanced results synthesis. Addressing all reviewers' specific comments, we also substantially improved the Mendelian Randomization (MR) and TWAS analyses. Of note, potential ancestry impact has now been investigated for both eQTL/TWAS and MR analyses.

We note that the primary goal of our study was conducting a GWAS of lung adenocarcinoma (LUAD) in East Asian (EA) populations with a substantial replication dataset. We assembled the largest dataset of lung adenocarcinoma cases in EA populations to date, identified multiple novel loci in EA populations, and used these findings to construct a new PRS specific to EA using a conservative approach that uses only SNPs that achieve genome-wide significance. As an added value to these main findings, we took advantage of readily available data in EUR populations to conduct initial cross-ancestry comparisons that, despite sample size imbalances, were able to make some important observations. However, more extensive analysis including developing an ancestry informed PRS and exploring other approaches to generating the PRS would make the manuscript less focused and is beyond its scope.

We could not disentangle genetic effects across histologies vs. across smoking patterns since our study was limited to only one histology, LUAD. A future study that included squamous cell carcinoma cases would be needed to disentangle the effects the reviewer refers to. Nevertheless, the heterogeneity we show between the PRS for never-smokers and smokers is fully valid for lung adenocarcinoma, which is by far the predominant histology among never-smokers, especially in EA populations.

In summary, we believe our manuscript will make an important contribution to the literature by substantially expanding the catalogue of established SNPs associated with risk of LUAD in EA populations, by showing potential functional implications through TWAS and colocalization analysis using for the first time a new and unique dataset of normal lung tissue eQTLs in an EA population, by laying the foundation and providing quantitative direction for the scale of future studies needed in this population to fully characterize its underlying genetic architecture, and making novel comparisons that provide for the first time evidence of heterogeneity in the underlying genetic architecture of lung adenocarcinoma in EA vs. EUR populations. The latter is admittedly not definitive, as we point out in the revised Discussion, but we believe it is an important first step.

Specific comments:

I suggest reconsidering the use of the term “trans” in this manuscript, please see PMID: 34741159 for discussion of the misuse of this term in genetic association studies. Multi-ancestry or cross-ancestry would be more appropriate.

Response: We appreciate the comment. We have modified “trans-ethnic meta-analysis” to “multi-ancestry meta-analysis” throughout the manuscript.

Please clarify if analyses were restricted to variants with $MAF \geq 0.01$ in both EAS and EUR populations, or if SNPs that were common in only one population were included.

Response: For multi-ancestry meta-analysis, only those variants with $MAF > 0.01$ in both populations were included. This has now been clarified in the main text: To identify variants shared by both EA and EUR populations, we performed a fixed effect, multi-ancestry GWAS meta-analysis including data from samples in EA (11,753 cases and 30,562 controls) and samples from EUR populations (11,273 cases and 55,483 controls) for variants with $MAF \geq 0.01$ in both populations.

The cross-ancestry comparisons seem limited and primarily based on genome-wide significance. Visual inspection of regional LD plots was also performed, but it’s not clear if this was systematically conducted for all variants. For example, what does “significantly different” mean here: “the majority of SNP associations initially identified in EA populations showed no evidence of association within EUR populations and most were significantly different.”

Response: We performed a systematic comparison for 38 risk variants that achieved genome-wide significance in either population, both populations, or multi-ancestry meta-analysis. The

first sentence is now revised as “We systematically compared the effect size in EA vs. EUR populations of 38 susceptibility variants for LUAD.” For the lead variant in each locus, we formally tested the effect size difference. Here, “significantly different” means ORs were statistically different between the two populations with $FDR \leq 0.05$. The sentence is now modified as “In contrast, out of the 20 SNP associations initially identified in EA populations, two had $MAF < 0.01$, 11 showed no evidence of association within EUR populations at $P < 0.05$ (Figures 3D and 3E, Supplementary Table 16), and 11 associations were significantly different between the two populations with $FDR < 0.05$.”

The section titled “Comparing the genetics of LUAD in EA and EUR populations” has vague statements, such as “majority of SNPs” or “several SNPs”. Please provide the specific numbers of variants and the denominator for each comparison.

Response: Following the suggestion, we revised the section to provide more detailed information:

“We systematically compared the effect size in EA vs. EUR populations of 38 susceptibility variants for LUAD. These included 12 variants identified in the current study, 26 variants previously reported in EA^{10,11,13-15,65} and/or EUR^{16,19,20} populations, and results of multi-ancestry meta-analyses combining data from EA and EUR²⁴ populations (Supplementary Table 16). As expected, ten SNP associations that were independently identified in both populations and through multi-ancestry analysis were very similar (Figures 3A, B, C). In contrast, out of the 20 SNP associations initially identified in EA populations, two had $MAF < 0.01$, 11 showed no evidence of association within EUR populations at $P < 0.05$ (Figures 3D and 3E, Supplementary Table 16), and 11 associations were significantly different between the two populations with $FDR < 0.05$. Similar population differences were observed among never-smokers and individuals with smoking history (Supplementary Figure 6). For variants with $MAF > 0.01$ in both populations, the lack of association in EUR populations did not seem to be driven by low MAF or lower statistical power, as $MAFs$ in both populations for most variants were similar and GWAS in both populations had adequate power to detect at least some evidence of association (Supplementary Table 17). Further, evaluation of gene region plots that spanned 500 kb for these loci within EUR populations showed no or very weak evidence of association for other variants in the region as well as the lead variants from the EA populations (Supplementary Figures 7A-I), with one exception (Supplementary Figure 7J). For 8 SNPs initially identified in EUR populations, there was evidence of association for 5 variants in EA populations (Figure 7F) although all variants were attenuated in the EA compared to the EUR population and one variant had MAF less than 1% in EA; moreover, two variants were significantly weaker (Supplementary Table 16, Supplementary Figure 8). Similar patterns were observed among never-smokers and individuals with smoking history (Supplementary Figure 6).”

The authors conclude that “The lack of association in EUR populations did not seem to be driven by low MAF or lower statistical power” – so then what could be driving these differences? Is it possible that there are different lead/index variants at some of these regions that were somehow not captured in the analysis?

Response: We appreciate the comment. There are multiple potential reasons that could have driven the differences, including (1) index variants may not have been well tagged, (2) different

exposures that may modify genetic effects, and (3) a different distribution of LUAD defined by driver mutations (Chen *et al.*, *Nature Genetics*, 2020) across populations that might have somewhat different genetic etiologies (consistent with our previous finding of GWAS variants displaying stronger association among cases that are *EGFR*-positive (Seow *et al.*, *Human Molecular Genetics*, 2016) and a recent study reporting the association between *EGFR* mutation frequency variation and genetic ancestry (Carrot-Zhang *et al.*, *Cancer Discovery*, 2021)). These are mentioned in Discussion:

“We systematically compared top GWAS findings that had been initially reported in either or both populations. After accounting for differences in MAFs and statistical power as well as the local LD pattern of each locus (500 kb each side of the lead variant), we found that a substantial number of the associations initially reported in EA populations showed no signal in EUR populations. This might either reflect causal variants for these loci not being tagged well in the EUR, or suggest important differences between EA and EUR in the genetic architecture of LUAD samples, which could be caused by differential environmental exposures. Finally, this observation is also consistent with distinct tumor molecular characteristics (e.g., *EGFR* mutation prevalence was higher in Asians than EUR populations) observed in LUAD, suggesting different etiologies influenced by genetic and/or environmental factors^{13,79,80}.”

POPCORN analysis of cross-population heterogeneity seems inconclusive and limited statistical power is given as the reason on page 14. This seems to contradict claims elsewhere in the paper. Or perhaps POPCORN is simply not the right method to use for looking at heterogeneity?

Response: We think that POPCORN is the right method for cross-population overall comparison (we estimated both the correlation of effect sizes and the correlation of explained variance of individual risk variants that accounted for different MAFs LD pattern in two populations) by appropriately using population specific reference genotype panels. We revised the text to explicitly state that the genetic correlation between never-smokers in EA and EUR populations was weaker than individuals with a smoking history (Line 416-418), suggesting the heterogeneity across populations was present primarily in never-smokers. In Discussion, we noted that a larger sample size would be needed to make definitive conclusions.

The criteria for novelty are not explicitly articulated. It would be helpful to distinguish novel loci from novel variants and provide some nuance around what is considered “novel” for a specific population or novel for LUAD histology. For instance, on page 9, none of the 5p15.33 variants appear to be truly novel: rs13167280 has been previously reported in several studies in European (PMID: 30059977; 28604730 22523397) and East Asian populations (PMID: 31326317) and rs62332591 has also been previously reported in Europeans (PMID: 28604730) and African Americans (PMID: 23221128). Furthermore, I wouldn’t characterize $R^2 < 0.25$ as low LD, particularly for detecting “novel” associations in such a well-characterized lung cancer susceptibility region as 5p15.33.

Response: We appreciate the comment. We now clarified this in Methods: “We consider the following variants as novel for the GWAS in EA: (1) the lead variant with $p < 5 \times 10^{-8}$ in a locus that has not been previously reported in either EA or EUR populations, or (2) a secondary variant with $p < 5 \times 10^{-8}$ conditioning on the lead variant in a previously reported locus in either EA or

EUR populations with the requirement that the LD $R^2 \leq 0.2$ between the secondary and the lead variants in both populations.”

We agree that it was inappropriate to make a claim for novelty at the 5p15.33 locus. For this locus, the conditional and joint analyses helped us nominate multiple variants that were independently associated with LUAD risk, although they had been reported previously as risk variants. Thus, we revised the text as: “Conditional analysis using GWAS summary statistics suggested two additional susceptibility variants rs13167280 (OR = 1.29, P = 4.07×10^{-13}) and rs62332591 (OR = 0.87, P = 3.21×10^{-8}), in the locus at 5p15.33 and one additional variant rs12664490 (OR = 0.81, P = 1.24×10^{-10}) in the locus at 6p21.1 (Table 3, Supplementary Fig. 2).”

Why was such an old version of GTEx used for tissue-specific analysis? It's possible that lack of enrichment signal could be due to the fact that GTEx v6 was used.

Response: We thank the reviewer for the suggestion. For the Stratified LD score regression (s-LDSC), our original analysis followed a published pipeline (<https://github.com/bulik/ldsc>) including extensive tissue-specific annotation datasets (Finucane, *et al. Nature Genetics*, 2018). These annotation datasets include tissue-specific expressed genes calculated in 53 tissue types from GTEx v6p and those calculated in 152 tissue types compiled from multiple published studies (Fehrmann *et al.*, *Nature Genetics* 2015; Pers *et al.*, *Nature Communications* 2015). The datasets also include tissue/cell-specific chromatin annotations from EnTEX (111 annotations in 26 tissue types) and Roadmap dataset (378 annotations in 85 tissue types).

For s-LDSC analysis of tissue-specific expressed genes, top 10% of the most differentially expressed genes for each tissue type are used to test heritability enrichment (Finucane, *et al. Nature Genetics*, 2018). Following the reviewer's suggestion, we re-calculated and compared tissue-specific expressed genes for both GTEx v8 (54 tissue types; 948 donors) and GTEx v6p (53 tissue types; 548 donors) datasets following the same analytic procedure (Finucane, *et al. Nature Genetics*, 2018). The t-statistics for testing differential expression were highly correlated between the V6p and V8 datasets (cor=0.96 for lung, median cor=0.97 for all 53 tissue comparisons). Furthermore, we observed that top 10% tissue-specific genes were highly similar between the two versions of data (a median of 83% of genes overlapped). For each of the 53 tissue types, we further tested enrichment using gene lists based on GTEx V8 and V6p data and found that enrichment p-values are similar. Because the final enrichment results need to combine GTEx and all the other above-listed datasets, we choose to report the original results based on the V6p data.

In fact, since this type of enrichment analysis only requires identifying the top 10% of tissue-specific genes, the results do not heavily depend on the sample size of the gene expression data. In their original paper, Finucane and colleagues described that s-LDSC approach can “identify significant enrichments even when the gene expression data have a small number of samples per tissue or cell type, in contrast to that with eQTL-based methods” (Finucane, *et al. Nature Genetics*, 2018). For example, the authors identified significant enrichments using a dataset with 2.8 samples per cell type on average.

The choice of 50 SNPs around the lead eQTL for eCAVIAR is not well motivated or consistent with other fine-mapping analyses in this manuscript. Please consider using a more standard, window-based approach, such as +/- 500kb. Similarly, the reasoning behind the extremely low threshold for the colocalization posterior probability (>0.01) is unclear. Please describe how this threshold was chosen and why this is appropriate for this analysis. Also, please distinguish between regional posterior probabilities and SNP-specific posterior probabilities.

Response: We thank the reviewer for raising this important point. The choice of +/-50 SNPs around the lead SNP in eCAVIAR analysis was originally based on the developer's recommendations (Hormozdiari *et al.*, *AJHG* 2016). Following the reviewer's suggestion, we further performed eCAVIAR analysis using a window of +/-100kb of the lead SNP, to be consistent with the window size we applied for HyPrColoc analysis. The window size is based on the reasoning that the LD blocks around the GWAS lead SNPs for the tested loci were mainly within this range. We now present the results based on this new analysis in Supplementary Table 10B and edited the text accordingly.

The cutoff of CLPP > 0.01 was also based on the developer's recommendations as applied in their publication (Hormozdiari *et al.*, *AJHG* 2016). To address the reviewer's comment, we further chose more comparable thresholding between eCAVIAR and HyPrColoc approaches to be consistent in declaring colocalization. The performances of eCAVIAR and HyPrColoc have been compared side-by-side using eCAVIAR CLPP > 0.01 and HyPrColoc posterior probability > 0.7 as respective cutoffs (Foley *et al. Nature Communications* 2021). We now present the colocalization results based on these consistent cutoffs in Supplementary Table 10B and edited the text accordingly.

Regional probability in HyPrColoc analysis is defined as the probability that all the traits (i.e., LUAD and gene expression) share an association with one or more variants within the region. This criterion includes the case of a single causal variant as well as the case of two traits having distinct causal variants in strong LD with each other. The final Posterior probability is based on meeting both regional association criterion and the second criterion ensuring a single shared putative causal variant (Foley *et al. Nature Communications* 2021). We now added the description of the terms in the footnote for Supplementary Table 10B.

Comparisons of the top vs. bottom decile should be avoided, for PRS and especially for MR. This presents a distorted view of the data that is only designed to elicit a large effect size by comparing two extremes. This is not an informative comparison because typically the goal of the PRS is to improve our ability to identify high-risk individuals in the general population, so comparisons should be with respect to the average risk group (40-60th percentile or 20-80th percentile). This is even more unnecessary for MR and hinders comparisons with other MR studies of lung cancer and TL. There is no need to go through the complicated conversion, just report the OR that corresponds to the increase in lung cancer risk per unit increase in telomere length and report what that "unit" corresponds to (which depends on how TL was operationalized in the TOPMed GWAS).

Response: We have revised the manuscript according to the reviewer's suggestion. For both PRS analyses, we now make comparisons with respect to the average risk group (40-60th percentile).

These results are shown in the updated Figure 4. For MR analysis, we report the OR per standard deviation of genetically predicted telomere length (Supplementary Table 15) and revised main text accordingly (Line 373-389).

Basic information about the MR analysis is missing and there are no MR diagnostics presented. Was the inverse variance weighted (IVW) analysis done using a fixed effects or multiplicative random effects model? Please clarify. Was there any evidence of heterogeneity in the SNP-specific causal effect estimates? Presence of heterogeneity would indicate potential for balanced horizontal pleiotropy, which can be addressed using appropriate methods (random effects IVW or MR PRESSO).

Response: We appreciate the reviewer's comment and suggestion. The original inverse variance weighted (IVW) analysis was based on fixed effects. Following the reviewer's comment, we now report the results based on MR-PRESSO (Verbanck *et al.*, *Nat. Genet.*, 2018) as the primary MR analysis, which handles horizontal pleiotropy automatically by identifying outliers and re-estimating causal effects after removing outliers. The estimated causal effects after removing these detected outliers are expected to be more robust. In fact, this method did identify 5-7 outlier variants in our analysis. We updated Methods (Line 679-695) and Results accordingly (Line 373-389).

Why were MR Egger, weighted median, and weighted mode chosen as additional MR methods? Was there any evidence of directional pleiotropy to justify the use of MR Egger? If the MR Egger intercept is not significantly different from zero, then there is no need to use MR Egger for estimating causal effects.

Response: We originally used these three methods for sensitivity analysis for potential existence of horizontal pleiotropy. We appreciate the suggestion of using MR-PRESSO, which identified outlier variants showing heterogeneity in the SNP-specific causal effect estimates. In the updated manuscript we report the estimates of causal effects based on the analysis after removing these identified outlier variants using MR-PRESSO. We deleted the results from these three methods.

MR analyses have a number of issues. Firstly, the instrument selection procedure is not described at all. How were these SNPs selected? What was the p-value and LD threshold? Given the differences in ancestry between the TOPMed telomere length GWAS and the present analysis, did the authors make any efforts to ensure that these SNPs and corresponding effect sizes were appropriate? Was there evidence of ancestry-related heterogeneity for any of these variants?

Response: We appreciate the comments and now described in greater detail about instrument selection in Methods. In particular, we chose variants that achieved genome-wide significance in TOPMED multi-ancestry meta-analysis of telomere length (Taub *et al.*, *Cell Genomics*, 2022). For loci with multiple variants, we kept variants with $LD R^2 < 0.05$ in both EUR and EA populations. See Methods (Line 680-689).

For the selected variants used as instruments, there was no evidence of heterogeneity in effect sizes across populations (Supplemental Table 4 in Taub *et al.*, *Cell Genomics*, 2022). Thus, in

the initial submission, we chose to use the effect sizes (on telomere length) estimated based on samples from all populations. Following the reviewer's comment, we have also performed analysis using effect sizes estimated based on Asian or EUR samples in TOPMed for MR analysis of LUAD in EA or EUR populations. Results were similar (Supplementary Table 15).

Weighted median and weighted mode are the same general class of MR estimator, these are consensus-based methods that assume that the true causal effect is the one estimated by the largest number of instruments. Including both is OK but a bit redundant. I would instead suggest including another MR estimation method that takes a different approach to pleiotropy detection and correction.

Response: We now use MR-PRESSO (Verbanck *et al.*, *Nat. Genet.*, 2018) for primary MR analysis.

*TWAS findings are not explored. For instance, is there eQTL sharing with other genes in the region, are *ELF5* and *FADS1* likely to be the main causal genes?*

Response: We thank the reviewer for this comment. Our TWAS analyses showed that *ELF5* and *FADS1* are the only transcriptome-wide significant genes associated with LUAD in 1Mb region of the respective GWAS signal; other gene models passing our performance criteria did not display significant TWAS P-values (*APIP* with $P = 0.0345$ in *ELF5* region; *CYBASC3* with $P = 0.0697$ and *INCENP* with $P = 0.969$ in *FADS1* region).

Although TWAS does not determine causality of gene expression for a trait, we performed a conditional analysis to assess whether the genetically predicted expression of *ELF5* and *FADS1* explain most of GWAS signal in the respective loci. For the locus around *ELF5*, when the GWAS signal was conditioned on the predicted expression of *ELF5* (represented by microarray probe ILMN_1684699 with the best TWAS p-value), most of the GWAS signal disappeared, and the TWAS signal for the second probe of *ELF5*, ILMN_1813270, was no longer significant (Supplementary Figure 4). For the locus around *FADS1*, when the GWAS signal was conditioned on the predicted expression of *FADS1* (represented by microarray probe ILMN_1670134), the strongest part of signal disappeared (Supplementary Figure 4). These data suggested that genetically predicted expression of *ELF5* and *FADS1* likely explain most or substantial part of GWAS signals in the respective loci. We now present these results in the manuscript: "When GWAS signal was conditioned on predicted expression of *ELF5*, most of the signal disappeared, adding support for *ELF5* as the main susceptibility gene in this locus (Supplementary Figure 4A)." "Similarly, when GWAS signal was conditioned on predicted expression of *FADS1*, the strongest part of the signal disappeared (Supplementary Figures 4). We further performed TWAS analysis using GTEx lung eQTL dataset (v8, $n = 515$, ~85% Europeans) and identified five genes from four loci (Supplementary Table 10). While identification of *ELF5* was common between two datasets, GTEx identified four unique genes from three known loci (*DCBLD1*, *MPZL3*, *JAML*, and *LINC00674*). Notably, *FADS1* was identified only by the ancestry-matched LCTCNS eQTL dataset even with a ~4 times smaller sample size."

The development of transcriptome imputation models using LCTCNS eQTL data is a strength of this paper, but it gets lost in everything else. Are the models publicly available? How do they compare with GTEx data in terms of the prediction performance and number of genes captured? What is the number of overlapping genes? Does TWAS using LCTCNS models identify genes that would not be detected using ancestry-discordant models?

Response: We appreciate this important comment; based on the reviewer's suggestion, we performed TWAS analysis using GTEx v8 lung eQTL data (n = 515) while incorporating LD matrix of 1000 Genomes EA populations. When we set a model performance criterion based on cross-validation R^2 ($R^2 > 0.05$; elastic net), 1,875 expression probes in LCTCNS dataset and 5,534 genes in GTEx dataset passed this cutoff. Using these numbers, transcriptome-wide significance cutoffs for TWAS P-values were set at $P < 2.7 \times 10^{-5}$ for LCTCNS (0.05/1875) and $P < 9 \times 10^{-6}$ for GTEx (0.05/5534). Based on these TWAS cutoffs, 3 probes for 2 genes (*ELF5* and *FADS1*) were significant in LCTCNS and 5 genes (*DCBLD1*, *MPZL3*, *JAML*, *LINC00674*, and *ELF5*) were significant in GTEx. While *ELF5* was overlapping between two datasets, the rest were unique to each dataset. Importantly, *FADS1* was not significant in the ancestry-discordant GTEx dataset, although GTEx had a > 4-fold larger sample size (115 vs. 515) and used a more powerful gene expression profiling method (microarray vs. RNA-sequencing). We now added the TWAS results based on GTEx dataset into the manuscript (Supplementary Table 11; Line 341-343).

LCTCNS expression data is publicly available through Gene Expression Omnibus (Accession No. GSE46539), and in-depth description of different expression models will be available in a separate manuscript.

The Discussion seems very brief and only focuses on selected findings, doesn't provide sufficient context for the other analyses and doesn't discuss any limitations or challenges with cross-ancestry comparisons (beyond sample size).

Response: Following the suggestion, we discussed the potential reasons for observed heterogeneity for susceptibility variants between EA and EUR populations, beyond minor allele frequencies, sample sizes, and local LD structure: "We systematically compared top GWAS findings that had been initially reported in either or both populations. After accounting for differences in MAFs and statistical power as well as the local LD pattern of each locus (500 kb each side of the lead variant), we found that a substantial number of the associations initially reported in EA populations showed no signal in EUR populations. It might either reflect causal variants for these loci not being tagged well in the EUR population, or suggest important differences between EA and EUR in the genetic architecture of LUAD samples, which could be caused by differential environmental exposures. Finally, this observation is also consistent with distinct tumor molecular characteristics (e.g., *EGFR* mutation prevalence was higher in Asians than EUR populations) observed in LUAD, suggesting different etiologies influenced by genetic and/or environmental factors^{13,79,80}."

In addition, we added a limitation paragraph, which discussed the challenges and limitations for cross-ancestry comparison as was suggested by the reviewer and other limitations commented by other reviewers: "There are several limitations in the current study. First, the discovery phase included subjects of diverse EA populations (Mainland China 38.2%, Japan 45.9%) and the

replication phase only included subjects from Japan. However, our data did not show evidence of heterogeneity in effect sizes for susceptibility variants between Han Chinese and Japanese populations or across geographic locations (Supplementary Table 5), suggesting a minimal impact for using a single EA population for replication. Second, we were underpowered to conduct formal heritability correlation analyses to compare the genetic architecture in EA and EUR populations stratified by smoking status; larger studies will be needed to conclusively characterize differences. Furthermore, completely elucidating the genetic basis of ancestry differences requires detailed information about age of onset, family history and exposures. Finally, rs4268071 (Table 2) achieved genome-wide significance in the discovery data but replication data were not available. While the significance was primarily driven by Japanese samples (MAF=0.04 in Japanese and <1% in other populations), there was no evidence of heterogeneity in effect estimates across EA populations. Replication is warranted to further establish its etiological role.”

Reviewer #3 (Remarks to the Author):

Shi and colleagues conducted a two-stage genome-wide association study of lung adenocarcinoma in individuals of East Asian (EA) ancestry, including 11,753 cases and 30,562 controls from four studies in the discovery phase and 9,905 cases and 120,114 controls from Japan in the replication phase, through which they identified 14 novel genetic risk variants. They performed post-GWAS functional characterization of the identified risk loci, encompassing eQTL colocalization and transcriptome-wide association analyses. In addition, they combined data from EA (11,753 cases and 30,562 controls) and European (EUR; 11,273 cases and 55,483 controls) populations in a trans-ethnic GWAS meta-analysis, identifying four genetic variants that confer similar risk effects for lung adenocarcinoma in both populations. They further compared risk effect sizes of both novel and previously known genetic risk variants for lung adenocarcinoma between EA and EUR populations. Additional analyses revealed that all known genetic variants combined as a polygenic risk score (PRS) and some individual genetic risk variants were more strongly associated with lung adenocarcinoma risk in individuals who never smoked than those who smoked; that about 2,275 genetic variants may be independently associated with lung adenocarcinoma in EA populations; and that expanding GWAS to include 70,000 cases and 70,000+ controls could further improve knowledge about the genetic architecture of lung adenocarcinoma, including genetic interactions with tobacco smoking and other environmental risk factors. Findings from their extensive work reinforce the importance of conducting genetic association studies of lung adenocarcinoma in non-EUR ancestral populations. Overall, the study rationale, methods, and results are clearly described and presented, and only minor comments are suggested to improve this well-written manuscript.

The discovery phase included individuals of diverse EA populations from mainland China, Hong Kong, Singapore, Taiwan, South Korea, and Japan, whereas the replication phase included individuals from Japan only. To what extent could differences in the composition of the discovery and replication sets have influenced the identification and confirmation of novel loci for lung adenocarcinoma, given that genetic differences do exist between EA ethnic groups? If this design choice was not ideal, it should be acknowledged as a study limitation.

Response: We appreciate the reviewer's insightful and constructive comment. We agree that it would be ideal to replicate associations in each ethnic-specific EA population. However, it was not logistically feasible to obtain a large replication set in each ethnic group whereas we were able to obtain data on a large number of independent cases and a very large number of controls from a major cohort study in Japan and felt that this would suffice given the similar genetic architectures of the populations we studied. Since majority of subjects in our study are of Chinese and Japanese origin, we compared the ORs for all 28 risk variants (including novel variants) between Han Chinese and Japanese, the two largest ancestry populations in the study. We found that $\log(\text{ORs})$ are highly correlated with $\text{cor}=0.98$ between the two populations. Furthermore, formal statistical testing did not reveal evidence of heterogeneity between the two populations ($P>0.1$ for all variants, Supplementary Table 5). These results strongly supported the conclusion of the current study and provided support that the replication data from the Japanese population was generalizable to the other ethnic groups. This point has been added to Discussion as a limitation.

Please clarify why no replication data were available for the variant identified on chromosome 7 (rs4268071). Of all the novel variants identified, this one is the least common (EAF=4%). Could this be a chance finding?

Response: When selecting variants for independent replication, we focused on variants with $\text{MAF} > 1\%$ across all sub-populations. While there were no replication data, there was no evidence of heterogeneity of effect sizes across populations based upon the GWAS data used for discovery, i.e., $\text{OR}=0.60$ ($\text{CI}=0.35-1.04$) for Han Chinese, $\text{OR}=0.74$ ($\text{CI}=0.67,0.82$) for Japanese, $p_{\text{het}} = 0.35$ using Cochran's Q statistic for testing heterogeneity), indicating that the variant is likely to be a true positive. Of note, the 95% CI for Chinese samples were especially wide because the variant is rare in that population. We agree with the reviewer that future replication would be critical to further establish its etiological role, which we have added to the Discussion as well as noting the limitation of our data with regard to this specific SNP.

3. For the PRS-smoking interaction analysis, the population examined does not appear to be specified. Please clarify whether this analysis was conducted on the EA population only or both the EA and EUR populations separately. Also, it would be informative if an explanation was offered as to why the PRS showed a significantly stronger association among those with no smoking history.

Response: The PRS-smoking interaction analysis was conducted only in EA populations, since the substantial amount of new data in this paper came only from EA populations. Our finding, together with our recent paper showing a stronger association of PRS for LUAD risk in non-coal users than in coal users (Blechter *et al.*, *Environment International*, 2021), provides evidence that genetic susceptibility may vary by exposure patterns in EA populations. The same pattern might be taking place in our data with regard to non-smokers vs. smokers; however, we think that more data will be needed before we can conclude that not only are there differences between never-smokers and individuals with a history of smoking, but that once the genetic architecture of LUAD is comprehensively characterized in EA populations with future expanded GWAS that ultimately never-smokers will be shown to have greater genetic risk. We prefer to use the conservative language in the current Discussion.

4. The text refers to Supplementary Table 8 only, rather than Supplementary Tables 8A, 8B, 8C, and 8D.

Response: We have modified the reference of the Supplementary Tables (now updated as Supplementary Table 9A, 9B, 9C, and 9D) accordingly.

5. Suggest reordering the presentation of the methods to follow that of the results.

Response: We have now reordered Methods according to Results.

6. Please use non-stigmatizing language (e.g., individuals with a smoking history, instead of smokers) whenever possible.

Response: Following the suggestion, we have modified “smokers” to “individuals with a history of smoking”.

REVIEWERS' COMMENTS

Reviewer #1 (Remarks to the Author):

No further comments, I am satisfied

Reviewer #2 (Remarks to the Author):

I really appreciate the thoughtful and extensive revisions performed by the authors, including new analyses and clarification of existing results. The authors conducted complementary colocalization analyses using HyPrColoc and present these alongside CAVIAR. The quality of the MR analysis is substantially improved, particularly with the addition of MR PRESSO and population-specific effect sizes for TL instruments. This is now a much more informative addition to the paper. TWAS conditional analyses are another great addition, as this helps prioritize the gene(s) most likely driving the signal within a locus. I commend the authors for their efforts and have no additional suggestions.

Reviewer #3 (Remarks to the Author):

I thank the authors for satisfactorily addressing my comments with detailed responses. The latest revisions, however, have raised a few additional comments that I would like the authors to address, which I believe will enhance clarity.

1. The authors now report identifying 12 novel susceptibility variants in EA populations overall (line 285). While Figure 1 presents 12 novel variants, Supplementary Table 4 presents 11 new variants achieving GWA in EA. Comparing Figure 1 and Supplementary Table 4, the remaining variant is in 14q13.2. Presumably, that corresponds to rs1200399; if so, why is this variant defined as novel but the other variant found in the multi-ancestry analysis is not? In addition, footnoting which variants were identified as novel in Supplementary Table 4 would be helpful.

At least to this reviewer, it becomes more confusing in the subsequent section on fine mapping and functional analyses, in which the 14 novel variants identified from the combined discovery and

replication datasets as well as conditional analysis were examined (line 307). Why are different subsets of variants being reported as novel?

2. With the PRS and gene-smoking interaction re-analysis, the magnitude of association for the top quintile does not appear "significantly higher" in individuals who have never smoked, relative to those who have smoked (OR=2.1 vs. OR=1.8). The authors may want to consider tempering this interpretation.

3. Line 278 should note Supplementary Figures 2A and 2B, not just Supplementary Figure 2. Lines 417-418 appear to pertain to Supplementary Table 13, instead of Supplementary Table 12.

We thank all the reviewers for the positive comments to our revision. We provide a point-by-point response to the specific comments from Reviewer #3 below in blue.

Reviewer #1 (Remarks to the Author):

No further comments, I am satisfied.

Reviewer #2 (Remarks to the Author):

I really appreciate the thoughtful and extensive revisions performed by the authors, including new analyses and clarification of existing results. The authors conducted complementary colocalization analyses using HyPrColoc and present these alongside CAVIAR. The quality of the MR analysis is substantially improved, particularly with the addition of MR PRESSO and population-specific effect sizes for TL instruments. This is now a much more informative addition to the paper. TWAS conditional analyses are another great addition, as this helps prioritize the gene(s) most likely driving the signal within a locus. I commend the authors for their efforts and have no additional suggestions.

We are very appreciative of the complementary comments from the reviewer.

Reviewer #3 (Remarks to the Author):

I thank the authors for satisfactorily addressing my comments with detailed responses. The latest revisions, however, have raised a few additional comments that I would like the authors to address, which I believe will enhance clarity.

1. The authors now report identifying 12 novel susceptibility variants in EA populations overall (line 285). While Figure 1 presents 12 novel variants, Supplementary Table 4 presents 11 new variants achieving GWA in EA. Comparing Figure 1 and Supplementary Table 4, the remaining variant is in 14q13.2. Presumably, that corresponds to rs1200399; if so, why is this variant defined as novel but the other variant found in the multi-ancestry analysis is not? In addition, footnoting which variants were identified as novel in Supplementary Table 4 would be helpful.

We appreciate the comment. We have modified Figure 1 and Supplementary Table 4 following the reviewer's suggestions. We identified 12 novel variants achieving genome-wide significance (GWS), including the lead variants for 10 novel loci, and two novel variants at previously reported loci (rs71467682 at 15q21.2, a locus previously reported in EUR populations; rs12664490 at 6p21.1, a locus previously reported in EA populations). The discrepancy between Figure 1 and Supplementary Table 4 in the last submission is rs12664490 in the locus at 6p21.1, not rs1200399 at 14q13.2. To improve clarity, we now list rs12664490 (conditional analysis at 6p21.1) in the category of "New variants achieving GWS in EA" in Supplementary Table 4, which includes all 12 novel variants achieving GWS. We put a footnote in Supplementary Table 4 to indicate that rs12664490 was based on a conditional analysis. We also edited the main text to improve clarity in counting the total number of novel variants identified in this study (Line 279-285).

At least to this reviewer, it becomes more confusing in the subsequent section on fine mapping and functional analyses, in which the 14 novel variants identified from the combined discovery and replication datasets as well as conditional analysis were examined (line 307). Why are different subsets of variants being reported as novel?

We apologize for this confusion. We now limited the variant annotation efforts to the 12 novel variants for consistency throughout the manuscript.

2. With the PRS and gene-smoking interaction re-analysis, the magnitude of association for the top quintile does not appear "significantly higher" in individuals who have never smoked, relative to those who have smoked (OR=2.1 vs. OR=1.8). The authors may want to consider tempering this interpretation.

We appreciate this comment. By "significantly higher", we meant that OR for the top quintile in never smokers was statistically significantly higher than that in individuals with a history of smoking ($p=0.0058$), when compared to the middle quantile. To avoid confusion, we modified the sentence as (Line 431-435):

“Compared to the middle quintile that represents the average risk in the general population, the top quintile had an OR of 2.07 (95% CI = 1.99, 2.15) for never-smokers and 1.80 (95% CI = 1.70, 1.89) for individuals with a history of smoking ($P_{\text{interaction}} = 0.0058$, Fig. 4, Supplementary Fig. 10), providing statistical evidence that the association between PRS and LUAD risk was higher for never-smokers.”

We note that in the original submission, we calculated the OR for each decile relative to the bottom decile (e.g., OR=5.79 for never-smokers and OR=4.15 for individuals with a history of smoking for the top 10th to the lowest 10th PRS decile). In the revision, we responded to the comment from the second reviewer by calculating the OR of each quintile relative to the middle quintile, which represents the average risk in the population (OR=2.07 for never-smokers, OR=1.80 for individuals with a history of smoking). The change reflects the different choice of the reference group for calculating OR and does not impact testing the statistical significance of the PRS*smoking interaction.

3. Line 278 should note Supplementary Figures 2A and 2B, not just Supplementary Figure 2. Lines 417-418 appear to pertain to Supplementary Table 13, instead of Supplementary Table 12.

Thank you for the comments. We have corrected them.